# VLM4Bio: A Benchmark Dataset to Evaluate Pretrained Vision-Language Models for Trait Discovery from Biological Images

**M. Maruf**[1*]     **Arka Daw**[2*]     **Kazi Sajeed Mehrab**[1]     **Harish Babu Manogaran**[1]
**Abhilash Neog**[1]     **Medha Sawhney**[1]     **Mridul Khurana**[1]     **James P. Balhoff**[3]
**Yasin Bakış**[4]     **Bahadir Altintas**[4]     **Matthew J Thompson**[5]     **Elizabeth G Campolongo**[5]
**Josef C. Uyeda**[1]     **Hilmar Lapp**[6]     **Henry L. Bart Jr.**[4]     **Paula M. Mabee**[7]
**Yu Su**[5]     **Wei-Lun Chao**[5]     **Charles Stewart**[8]     **Tanya Berger-Wolf**[5]
**Wasila Dahdul**[9]     **Anuj Karpatne**[1]

[1]Virginia Tech     [2]Oak Ridge National Laboratory     [3]UNC Chapel Hill
[4]Tulane University     [5]Ohio State University     [6]Duke University     [7]Battelle
[8]Rensselaer Polytechnic Institute     [9]UC Irvine
{marufm,darka,karpatne}@vt.edu

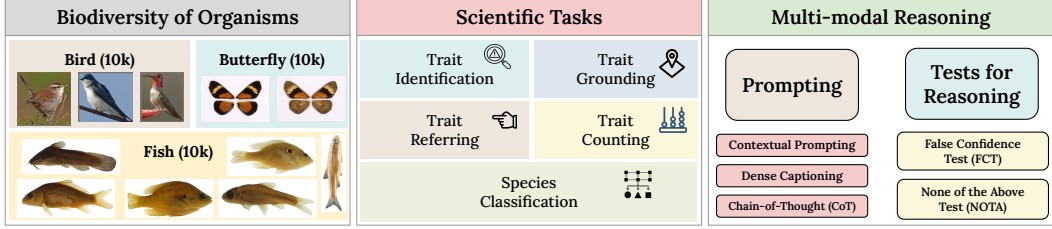

Figure 1: Overview of our goals and contributions. We analyze the capabilities of 12 state-of-the-art (SOTA) vision-language models (VLMs) in answering scientific questions using images from three groups of organisms: fishes, birds, and butterflies, over five groups of biologically relevant tasks. We also explore the effectiveness of these models for reasoning using various prompting techniques and tests for reasoning hallucination.

## Abstract

Images are increasingly becoming the currency for documenting biodiversity on the planet, providing novel opportunities for accelerating scientific discoveries in the field of organismal biology, especially with the advent of large vision-language models (VLMs). We ask if pre-trained VLMs can aid scientists in answering a range of biologically relevant questions without any additional fine-tuning. In this paper, we evaluate the effectiveness of 12 state-of-the-art (SOTA) VLMs in the field of organismal biology using a novel dataset, **VLM4Bio**, consisting of $469K$ question-answer pairs involving $30K$ images from three groups of organisms: fishes, birds, and butterflies, covering five biologically relevant tasks. We also explore the effects of applying prompting techniques and tests for reasoning hallucination on the performance of VLMs, shedding new light on the capabilities of current SOTA VLMs in answering biologically relevant questions using images. [1]

---

[1]The code and datasets for running all the analyses reported in this paper can be found at https://github.com/imageomics/VLM4Bio.

38th Conference on Neural Information Processing Systems (NeurIPS 2024) Track on Datasets and Benchmarks.

# 1  Introduction

There is a growing deluge of images that are being collected, stored, and shared in organismal biology—the branch of biology interested in the study of structure, ecology, and evolution of organisms. In particular, images are increasingly becoming the currency for documenting the vast array of biodiverse organisms on our planet, with repositories containing millions of images of biological specimens collected by scientists in field museums or captured by drones, camera traps, or tourists posting photos on social media. This growing wealth of biological images provides a unique opportunity to understand the scientific mechanisms of how organisms evolve and adapt to their environment directly from images. The traditional approach for advancing knowledge in organismal biology is by discovering the observable characteristics of organisms or *traits* (e.g., beak color, stripe pattern, and fin curvature) that serve a variety of biological tasks such as defining groups of organisms, understanding their genetic and developmental underpinnings, and analyzing their interactions with environmental selection pressures [1]. However, the measurement of traits is not straightforward and often relies on expert visual attention involving labor-intensive operations and subjective definitions [2], hindering rapid scientific advancement [3].

With the recent rise of large foundation models such as vision-language models (VLMs) (e.g., GPT-4, GPT-4V(ision) [4, 5], Gemini [6], LLaMA 3.2 [7, 8], and LLaVA [9]) that can simultaneously solve a diverse range of tasks involving text and images, it is pertinent to ask if pre-trained VLMs contain the necessary *scientific knowledge* to aid biologists in answering a variety of questions pertinent to the discovery of biological traits from images. Note that unlike mainstream tasks in computer vision, understanding scientific images requires knowledge of domain-specific terminologies and reasoning capabilities that are not fully represented in conventional image datasets used for training VLMs. In particular, an important end-goal in scientific applications such as organismal biology is to explain the process of visual reasoning used to arrive at a prediction, often involving the knowledge of biological traits. Hence, to assess the usefulness of VLMs in accelerating discoveries in organismal biology, it is important to test their ability to identify and reason about biological traits automatically from images.

In this work, we assess the zero-shot capabilities of 12 state-of-the-art (SOTA) VLMs, including the proprietary GPT-4V(ision) and the recent GPT-4O(mni) along with other open-source VLMs, on five scientifically relevant tasks in organismal biology, namely species classification, trait identification, trait grounding, trait referring, and trait counting. These tasks are designed to test different facets of VLM performance in organismal biology, ranging from measuring predictive accuracy to assessing their ability to reason about their predictions using visual cues of known biological traits. For example, the task of species classification tests the ability of VLMs to discriminate between species, while in trait grounding and referring, we specifically test if VLMs are able to localize morphological traits (e.g., the presence of fins or patterns and colors of birds) within the image. To perform this evaluation, we present **VLM4Bio**, a benchmark dataset of $\approx 469K$ question-answer pairs based on $30k$ images of three taxonomic groups of organisms: fishes, birds, and butterflies.

**Main Contributions:**

1. We present a novel dataset of scientific question-answer pairs to evaluate the effectiveness of VLMs in answering scientific questions across a range of biologically relevant tasks in the field of organismal biology.

2. We present novel benchmarking analyses of the zero-shot effectiveness of pre-trained SOTA VLMs on our dataset, exposing their gaps in advancing scientific knowledge of organismal biology.

3. We present novel comparisons studying the effects of prompting and tests for reasoning hallucination on VLM performance, shedding new light on the reasoning capabilities of SOTA VLMs in organismal biology.

# 2  Related Works

With the rise of SOTA VLMs such as GPT-4V(ision) [5], GPT-4O(mni) [10], and Gemini [6], there has been a simultaneous growth in the number of benchmarking analyses published in the last few years to evaluate different facets of VLM performance on a range of mainstream tasks in computer vision. A majority of previous analyses [11, 12] involve evaluations on single tasks like Visual

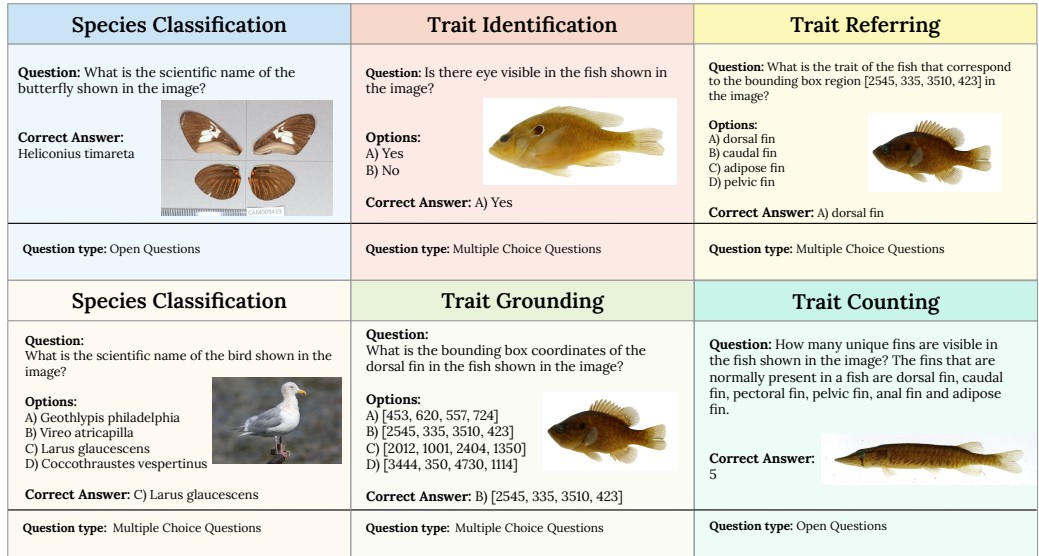

Figure 2: Illustrative examples of **VLM4Bio** tasks with different question-types.

Question Answering (VQA), OK-VQA [13], MSCOCO [14], and GQA [15]. Other datasets such as POPE [16], HaELM [8], LAMM [17], MMBench [18], MM-Vet [19], LVLM-eHub [20], SEED [21], and GAIA [22] have also been developed to evaluate the capabilities of VLMs on complex tasks such as reasoning and ability to handle multimodal data. There are also some recent domain-specific benchmark datasets, such as MathVista [23], which includes a variety of challenging VQA problems in the mathematical domain, MedQA(USMLE) [24] which is a collection of VQA problems from medical exams, and the recent MMMU [12] dataset, which covers expert-level problems from diverse fields such as business, arts, science, health, medicine, and engineering.

VLM4Bio dataset is different from existing benchmarks involving domain-specific datasets because of the following reasons. (1) *Focus on organismal biology*: While previous works have focused on benchmarking the performance of VLMs on other scientific domains (e.g., Arts and Design, Business, Health, and Medicine in MMMU [12] or Mathematics in MathVista [23]), there exists no previous VQA benchmark dataset in the domain of organismal biology to the best of our knowledge. Our work thus fills a critical gap in evaluating the performance of VLMs in a field of biology that has several societal implications such as monitoring biodiversity and understanding the impact of climate change on species traits and populations. (2) *Breadth of Evaluation Tasks*: While previous works are tailored to one or a few evaluation tasks, we consider a wide range of tasks motivated by the needs of domain scientists in the field of organismal biology. They include predictive tasks such as species classification and trait identification as well as tasks that require visual reasoning including trait grounding and referring. We also provide novel comparisons about the performance of VLMs on both open-ended and multiple-choice question (MCQ) formats and comparisons over predictive as well as visual reasoning tasks, in contrast to prior works.

## 3 VLM4Bio Tasks

Figure 2 shows illustrative examples of the five VLM4Bio tasks relevant to biologists that we consider in our study, described in detail in the following.

### 3.1 Species Classification

A common (and often the first) task that a biologist considers when examining an organism specimen is to identify its scientific name (or species class). Hence, we consider asking a VLM to provide the scientific name of the organism shown in a given image. There are two types of questions that we consider for this task. First, we consider *open-ended questions*, where we do not provide any answer choices (or options) to the VLM in the input prompt. The second type is *multiple-choice*

*(MC) questions*, where we provide four choices of candidate species names for the VLM to choose from (out of which only one is correct while the remaining three are randomly selected from the set of all species classes).

### 3.2 Trait Identification

An important goal in organismal biology is to answer questions regarding the observable characteristics of organisms, also known as traits. We thus consider asking VLMs to identify a particular trait of an organism given its image for two taxonomic groups: fishes and birds. For fishes, we considered 10 binary (presence/absence) traits and generated MC questions for the presence of each trait in an image (with two options: yes or no), whereas for birds, we considered 28 traits covering their color, pattern, and measurements (size and shape of regions) in a multiple-choice format. We provide a detailed list of all fish and bird traits in the Supplementary Section F.

### 3.3 Trait Grounding and Referring

To further understand the ability of VLMs to visually explain the reasoning behind their prediction of a trait, it is important to evaluate if a VLM correctly identifies the region in the image containing the trait. For this purpose, we consider two other tasks: trait grounding & trait referring, for the taxonomic groups of fishes and birds. In the first task of trait grounding, we ask the VLM to locate a given trait of an organism on its image (i.e., *text to location*). We consider MC question-format for this task where we provide four options of bounding boxes in the image as candidate answer choices, where one of the bounding boxes correctly contains the trait while the remaining three are randomly sampled from the set of bounding boxes containing other traits of the organism. In the second task of trait referring, we consider the opposite scenario where we provide a bounding box as input to the VLM and ask it to identify the name of the trait present in the bounding box (i.e., *location to text*). We again provide four answer choices in MC question-format, where only one of the options is correct while the remaining three are randomly sampled from the names of other traits of the organism.

### 3.4 Trait Counting

We simply ask how many traits are present in an image of a fish specimen. This is biologically relevant, for example, to understand the number of fins present in a fish organism. Similar to the species classification task, we have open and MC question-types for this task.

## 4 VLM4Bio Dataset

**Data Collection and Preprocessing**: We collected images of three taxonomic groups of organisms: fish, birds, and butterflies, each containing around $10K$ images. Images for fish (**Fish-10K**) were curated from the larger image collection, FishAIR [25], which contains images from the Great Lakes Invasive Network Project (GLIN) [26]. These images originate from various museum collections such as INHS [27], FMNH [28], OSUM [29], JFBM [30], UMMZ [31] and UWZM [32]. We created the Fish-10K dataset by randomly sampling $10K$ images and preprocessing the images to crop and remove the background. For consistency, we leverage GroundingDINO [33] to crop the fish body from the background and Segment Anything Model (SAM) [34] to remove the background. We curated the images for butterflies (**Butterfly-10K**) from the Jiggins Heliconius Collection dataset [35], which has images collected from various sources [2]. We carefully sampled $10K$ images for Butterfly-10K from the entire collection to ensure the images capture unique specimens and represent a diverse set of species by adopting the following two steps. First, we filter out images with more than one image from the same view (i.e., dorsal or ventral). Second, we ensure each species has a minimum of 20 images and no more than $2,000$ images. The images for birds (**Bird-10K**) are obtained from the CUB-200-2011 [61] dataset by taking 190 species for which the common name to scientific name mapping is available. This results in a fairly balanced dataset with around $11K$ images in total. Additional details on dataset preprocessing are provided in the Supplementary Section A.

**Annotation:** The scientific names for the images of Fish-10K and Butterfly-10K were obtained directly from their respective sources. For Bird-10K, we obtained the scientific names from the

---

[2]Sources: [36, 37, 38, 39, 40, 41, 42, 43, 44, 45, 46, 47, 48, 49, 50, 51, 52, 53, 54, 55, 56, 57, 58, 59, 60]

| Statistics | **Fish-10K** | **Bird-10K** | **Butterfly-10K** | **Fish-500** | **Bird-500** |
|---|---|---|---|---|---|
| # Images | 10,347 | 11,092 | 10,013 | 500 | 492 |
| # Species | 495 | 188 | 60 | 60 | 47 |
| # Genera | 178 | 114 | 27 | 18 | 33 |
| # Traits | 10 | 28 | - | 8 | 5 |

Table 1: Key statistics of the **VLM4Bio** dataset.

iNatLoc500 [62] dataset. We curated around $31K$ question-answer pairs in both open and multiple-choice (MC) question formats for evaluating species classification tasks. The species-level trait presence/absence matrix for Fish-10K was manually curated with the help of biological experts co-authored in this paper. We leveraged the Phenoscape knowledge [63] base with manual annotations to procure the presence-absence trait matrix. For Bird-10K, we obtained the trait matrix from the attribute annotations provided along with CUB-200-2011. We constructed approximately $380K$ question-answer pairs for trait identification tasks. For grounding and referring VQA tasks, the ground truths were manually annotated with the help of expert biologists on our team. We manually annotated bounding boxes corresponding to the traits of 500 fish specimens and 500 bird specimens, which are subsets of the larger Fish-10K and Bird-10K datasets, respectively. In particular, we considered 8 fish traits and 5 bird traits for annotating their bounding boxes, resulting in a total of $26K$ question-answer pairs. We also used the Fish-500 dataset for the task of trait counting, resulting in a total of $1K$ question-answer pairs. Across all tasks, our dataset comprises approximately $469K$ question-answer pairs for $30K$ biological images (see Table 1). Additional details on data distribution and key statistics are provided in the Supplementary Section E.

**Dataset Card:** We provide the dataset card with a detailed description of the metadata, data instances, annotation, and license information here (`https://huggingface.co/datasets/imageomics/VLM4Bio#dataset-card-for-vlm4bio`).

**VLM Baselines:** We consider the following VLM baselines: GPT-4V(ision) [64][3], LLaVA-v1.5 (7B/13B) [65], COG-VLM [66], MiniGPT-4 (Vicuna 7B/13B) [67], BLIP-FLAN-T5-XL/XXL [68], and INSTRUCT-BLIP (Vicuna 7B/13B) [69]. We used the latest checkpoints for each model available to date. We used the same question prompt for all models to ensure consistent comparison of results for a variety of open and multiple-choice (MC) questions across the five scientific tasks of our dataset. All the experiments were conducted using NVIDIA A100 GPUs. See supplementary Section H for more details of the VLM baselines.

**Evaluation Metrics:** We used micro-averaged accuracy as our evaluation metric for all experiments. We designed a systematic rule-based evaluation pipeline to evaluate VLM responses against the ground truths. For each question category, we provide the accuracy percentage of random choice as a basic baseline, where each possible answer is considered equally likely (yielding an accuracy of 25% for MC questions with four choices).

## 5 Results

Table 2 compares the accuracies of VLMs in percentages (ranging from 0 to 100) across the five tasks and over multiple organism datasets. We make the following observations from this result.

**All VLMs show poor accuracy on open questions but perform better on MC questions.** The zero-shot species classification accuracy of all VLMs on open-ended questions is notably weaker than MC questions. Even the best-performing models, LLaVA-13B, GPT-4V, and Instruct-Vicuna-7B, only achieve accuracies of 2.32%, 17.46%, and 3.62%, respectively, across the three organism datasets. This indicates a significant gap in the ability of existing VLMs to capture the scientific knowledge necessary to differentiate between species (often requiring subtle or nuanced features) without being provided with candidate answer choices. Open-ended species classification is particularly hard for pre-trained VLMs that are not typically trained to provide scientific names of organisms (e.g., *Lepomis cyanellus*) rather than providing their common names (e.g., *green sunfish*). However, the inclusion of candidate answers (or options) in the question prompt serves as a helpful clue to VLMs

---

[3]We use *gpt-4-1106-vision-preview* model as GPT-4V in our experiments.

| Dataset | Question type | Models | | | | | | | | | | | | |
|---|---|---|---|---|---|---|---|---|---|---|---|---|---|---|
| | | gpt-4v | llava v1.5-7b | llava v1.5-13b | cogvlm chat | BLIP flan-xl | BLIP flan-xxl | minigpt4 vicuna-7B | minigpt4 vicuna-13B | instruct flant5xl | instruct flant5xxl | instruct vicuna7B | instruct vicuna13B | Random Choice |
| **Species Classification** | | | | | | | | | | | | | | |
| **Fish-10k** | Open | 1.01 | 2.32 | 0.40 | 0.11 | 0.01 | 1.59 | 0.50 | 0.38 | 0.00 | 1.46 | 0.00 | 0.00 | 0.20 |
| | MC | 35.91 | 40.20 | 32.27 | 31.72 | 29.76 | 33.36 | 29.02 | 27.45 | 30.86 | 31.70 | 27.27 | 26.57 | 25.00 |
| **Bird-10k** | Open | 17.40 | 1.45 | 2.06 | 0.86 | 0.00 | 0.57 | 2.80 | 2.56 | 0.00 | 0.50 | 0.07 | 0.00 | 0.53 |
| | MC | 82.58 | 50.32 | 55.36 | 44.73 | 33.68 | 34.75 | 23.95 | 27.62 | 36.36 | 35.83 | 44.00 | 46.55 | 25.00 |
| **Butterfly-10k** | Open | 0.04 | 0.05 | 0.00 | 0.01 | 0.00 | 0.00 | 0.07 | 0.01 | 0.00 | 0.00 | 9.94 | 0.00 | 1.54 |
| | MC | 28.91 | 50.24 | 44.58 | 36.45 | 25.14 | 28.88 | 33.06 | 28.90 | 25.28 | 36.67 | 41.70 | 34.48 | 25.00 |
| **Trait Identification** | | | | | | | | | | | | | | |
| **Fish-10k** | MC | 82.18 | 56.84 | 45.15 | 46.92 | 68.36 | 39.33 | 55.08 | 51.87 | 64.34 | 39.26 | 81.95 | 20.69 | 50.0 |
| **Bird-10k** | MC | 62.22 | 34.68 | 46.14 | 63.93 | 50.11 | 41.38 | 39.11 | 40.44 | 47.89 | 45.52 | 77.91 | 89.98 | 31.12 |
| **Trait Grounding** | | | | | | | | | | | | | | |
| **Fish-500** | MC | 29.41 | 24.87 | 17.98 | 23.42 | 23.32 | 25.14 | 22.18 | 25.58 | 7.20 | 27.09 | 33.51 | 26.90 | 25.00 |
| **Bird-500** | MC | 8.1 | 26.92 | 35.36 | 23.2 | 11.83 | 10.52 | 15.39 | 24.22 | 3.48 | 0.81 | 30.24 | 13.91 | 25.00 |
| **Trait Referring** | | | | | | | | | | | | | | |
| **Fish-500** | MC | 28.15 | 27.07 | 29.14 | 28.19 | 24.93 | 25.68 | 39.24 | 31.21 | 31.75 | 25.78 | 28.04 | 32.73 | 25.00 |
| **Bird-500** | MC | 42.28 | 30.5 | 29.64 | 18.45 | 35.16 | 40.59 | 26.04 | 35.88 | 27.52 | 41.69 | 23.03 | 22.69 | 25.00 |
| **Trait Counting** | | | | | | | | | | | | | | |
| **Fish-500** | Open | 16.4 | 47.4 | 52.0 | 14.8 | 37.6 | 63.4 | 13.6 | 31.53 | 50.2 | 61.4 | 61.4 | 0.0 | 25.00 |
| | MC | 44.80 | 13.20 | 54.80 | 21.00 | 64.8 | 78.2 | 22.00 | 25.00 | 74.0 | 69.4 | 15.80 | 11.80 | 25.00 |
| *Overall* | | 34.24 | 29.0 | 31.78 | 25.27 | 28.91 | 30.24 | 23.0 | 25.19 | 28.49 | 29.79 | 33.92 | 23.31 | 22.03 |

Table 2: Zero-shot accuracy comparison of VLM baselines (in % ranging from 0 to 100) for the five scientific tasks. Results are color-coded as Best , Second best , Worst , Second worst .

for narrowing down the solution space and finding the correct answer potentially using elimination strategies. While VLMs are able to utilize these additional hints and work their way through to the correct answer in MC questions, note that open questions are practically more relevant to scientists operating in real-world settings.

**Bird dataset shows better accuracy than Fish or Butterfly datasets.** Most VLMs show significantly better performance on the Bird-10K dataset in comparison to the Fish-10K and Butterfly-10K datasets. For example, the highest accuracy across all VLMs on the Bird-10K dataset is 82.58%, while it is 40.20% and 50.24% on the Fish-10K and Butterfly-10K datasets, respectively. A potential reason is that while the bird dataset is a subset of the CUB dataset [70] that is commonly used in machine learning literature and has images with natural in-the-wild backgrounds, the butterfly and fish datasets contain images of specimens preserved in museum collections with artificial backgrounds and with imaging artifacts that are not typical for large-scale computer vision datasets. We hypothesize that many of the pre-trained VLM baselines may have seen images similar to those in the Bird dataset during training, leading to their better performance.

**Can VLMs effectively identify biological traits?** The performance of most VLMs in trait identification appears significantly better than their performance in species classification, with GPT-4V reaching 82.18% accuracy on the Fish-10K dataset and Instruct-Vicuna-13B achieving 89.98% on Bird-10K. However, some traits such as "eye", "head", and "mouth" are almost always present in every organism image, so simply answering "yes, the trait is present" can lead to high accuracy in trait identification. In contrast to the fish dataset, the bird dataset poses more intricate questions regarding a variety of multi-class traits that require a nuanced understanding of colors, patterns, and physical trait dimensions, such as the color of the bill, wing patterns, and tail shapes.

**VLMs struggle in localizing traits in images.** While most VLMs perform well on the task of Trait Identification, it is crucial to determine if they are focusing on the correct image regions to answer trait-related questions. We thus analyze the performance of VLMs on the tasks of trait grounding (i.e., *text to location*) and trait referring (i.e., *location to text*). We can see that there is a significant

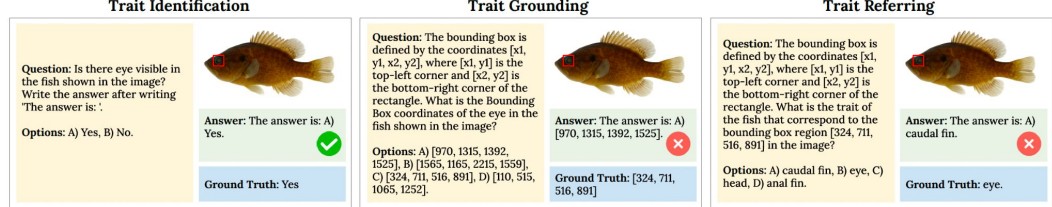

Figure 3: Examples of correct and incorrect predictions of GPT-4V for trait identification, trait grounding, and trait-referring tasks related to the "eye". For visualization assistance, a red-colored bounding box is added around the "eye" in the image.

drop in the accuracy of trait grounding and referring tasks compared to the trait identification task. This shows that while VLMs can potentially leverage knowledge of trait choices to identify traits, they struggle in localizing the traits in the image and thus visually ground their reasoning. Figure 3 shows an illustrative example of GPT-4V prediction where it predicts the presence of the trait "eye" correctly but fails to localize it in grounding and referring tasks.

**Counting biological traits is difficult for VLMs.** Recent studies [71, 72, 73] have explored the gap in the ability of VLMs to count objects, which is aligned with our results in Table 2. All VLMs, except for BLIP-flan-T5-XXL, show lower performance in counting traits, despite performing well on the trait identification task. The overall average accuracy for the VLMs is displayed in the last block, with GPT-4V(ision) exhibiting the best performance.

We further analyze the errors of different VLMs to better understand their behavior. We find that GPT-4V shows a reduced rate of incorrect responses but a higher incidence of "Other" responses, which include apologetic expressions, admissions of inability to precisely visualize the organism, and disclaimers regarding lack of expert guidance (see Supplementary Section J for more details).

### 5.1 Analyzing the Role of Answer Choices in MC Questions on VLM Performance

Table 2 showed that VLMs perform drastically better on MC questions compared to Open questions for species classification. A potential hypothesis for this observation is that VLMs are able to avoid incorrect answer choices (or options) that are too different from the correct option and thus are easy to eliminate. To test this hypothesis, we create three variants of the MC questions for species classification—easy, medium, and hard—-where species choices in each variant have varying degrees of similarity determined by their taxonomic groupings. In particular, note that the scientific name of an organism contains taxonomic information at three levels: `<genus name>` `<species name>` `<subspecies name>`[4]. Since organisms that share taxonomic information have similar appearances, it is hard to differentiate species choices if they are from the same taxonomic group. On the other hand, it is easier to work with species choices from different taxonomic groups. Hence, for the easy set, we selected 50 species from different genera, ensuring that all species choices appear quite different from each other. For the medium set, we increased the complexity by constructing species choices from the same genus but from 10 different species. The hard set presented the highest difficulty level for the butterfly dataset, with the answer choices being from the same genus and species but from 10 subspecies. Each difficulty level consists of 200 images from each set of organisms.

Table 3 shows the accuracies of the baseline VLMs for the easy, medium, and hard organism datasets. The pretrained VLMs generally perform best on the easy set and worst on the hard set for each organism. Moreover, there is a gradual improvement in the VLM performance from hard to easy questions. This suggests that the difficulty level of candidate answers (or options) in the question prompt significantly impacts VLMs' performance. Additionally, this outcome indicates that even SOTA VLMs have limitations in handling fine-grained queries. Table 3 shows that GPT-4V and OpenAI's recent release GPT-4o do not perform well when tested on the medium and hard datasets for Fish and Butterfly. Due to this, we further analyze the errors of different VLMs to better understand their behavior. We provide the report in the Supplementary Section J.

---

[4]We only have subspecies level information for the Butterfly-10K dataset.

| | | Models | | | | | | | | | | | | | | |
|---|---|---|---|---|---|---|---|---|---|---|---|---|---|---|---|---|
| **Dataset** | **Difficulty** | *gpt-4v* | *gpt-4o* | *llava v1.5-7b* | *llava v1.5-13b* | *cogvlm chat* | *BLIP flan-xl* | *BLIP flan-xxl* | *minigpt4 vicuna-7B* | *minigpt4 vicuna-13B* | *instruct flant5xl* | *instruct flant5xxl* | *instruct vicuna7B* | *instruct vicuna13B* | *CLIP* | *BioCLIP* |
| **Fish** | Easy | 44.50 | 37.50 | 47.50 | 46.00 | 24.00 | 34.00 | 27.50 | 29.00 | 19.50 | 32.00 | 28.00 | 33.50 | 33.50 | 36.50 | 55.50 |
| | Medium | 3.50 | 5.50 | 30.00 | 28.50 | 27.00 | 26.00 | 23.00 | 26.50 | 25.00 | 28.50 | 24.50 | 26.00 | 25.50 | 26.00 | 29.00 |
| **Bird** | Easy | 73.50 | 68.00 | 53.50 | 50.00 | 38.50 | 34.50 | 36.00 | 21.00 | 32.00 | 41.00 | 33.00 | 43.50 | 39.00 | 57.00 | 94.00 |
| | Medium | 41.00 | 40.50 | 30.50 | 37.00 | 30.00 | 25.50 | 21.00 | 21.00 | 24.00 | 27.00 | 27.00 | 24.50 | 26.50 | 31.00 | 95.00 |
| **Butterfly** | Easy | 18.50 | 17.50 | 19.00 | 20.50 | 24.50 | 30.00 | 25.00 | 34.50 | 26.00 | 24.50 | 22.50 | 19.00 | 24.50 | 21.50 | 65.50 |
| | Medium | 5.50 | 7.00 | 29.50 | 29.00 | 29.50 | 20.00 | 25.50 | 33.00 | 25.00 | 27.50 | 25.00 | 25.00 | 25.00 | 21.50 | 58.00 |
| | Hard | 2.00 | 1.50 | 22.00 | 21.00 | 32.00 | 26.50 | 20.00 | 29.50 | 24.00 | 22.50 | 24.00 | 24.00 | 21.00 | 21.50 | 35.00 |

Table 3: Zero-Shot accuracy comparison for *easy, medium, and hard* datasets. Results are color-coded as Best , Second best , Worst , Second worst .

## 5.2 Comparing Pre-trained VLMs with a Biologically Fine-tuned Model

We compare BioCLIP [74], a state-of-the-art foundation model for species classification fine-tuned with biological images and taxonomic names (TreeOfLife-10M dataset), with the pretrained VLMs. We observe that BioCLIP significantly outperforms large pretrained VLMs on the Bird-10k and Butterfly datasets, suggesting that BioCLIP has been trained on images that are similar to the organisms present in these datasets. By comparing BioCLIP with CLIP, we can also see that fine-tuning foundation models with biological data provides large gains in classification performance. This suggests that the performance of SOTA VLMs can be further improved by fine-tuning on VLM4Bio Dataset. Further details comparing BioCLIP with SOTA VLMs are provided in the Supplementary Section K.

## 5.3 Analyzing Effects of Prompting on VLM Performance

We considered three prompting techniques: Contextual Prompting, Dense Caption Prompting, and zero-shot Chain of Thought Prompting. For **Contextual prompting**, we provided a single-line description (context) of the tasks (e.g., we add "*Each biological species has a unique scientific name composed of two parts: the first for the genus and the second for the species within that genus.*" before the species classification question to give some additional context on the task). **Dense Caption prompting** involves two stages: (1) first, we prompt the VLM to generate a dense caption of the specimen image such that the caption contains all the necessary trait information of the specimen. (2) We add the dense caption before the question and prompt "*Use the above dense caption and the image to answer the following question.*" to generate responses from the VLM. Similarly, the **Zero-Shot Chain-of-Thought (CoT)** happens in two stages: (1) first, we prompt the VLM to generate the reasoning for a given VQA and multiple choices (options). Zero-shot CoT appends "*Let's think step by step.*" after the question and options to generate the reasoning. (2) We then add the reasoning after the VQA and prompt "*Please consider the following reasoning to formulate your answer*" to generate the VLM response. We curated a prompting dataset of 500 multiple-choice (MC) VQAs for each set of organisms, which is a subset of the VLM4Bio dataset for species classification.

Table 4 compares best-performing VLMs on the prompting dataset. The CoT rows of the table demonstrate that only GPT-4V and GPT-4o have reasoning capabilities that can significantly improve their response to biological questions, while smaller models like LLaVa and BLIP do not show much improvement. Furthermore, providing extra context and caption is more useful for GPT-4V and GPT-4o than the smaller models. This resonates with the findings from [75] that the reasoning abilities of VLMs only emerge after a certain model size. The success of Dense Caption Prompting and CoT Prompting depends on how well they generate the dense caption or the reasoning in the first stage. We report example prompts with VLM responses as case studies in the Supplementary Section M.

## 5.4 Analyzing Tests for Reasoning Hallucination

To further understand whether pretrained VLMs can respond with logically coherent and factually accurate reasoning, we evaluate VLMs on two sets of reasoning for hallucination tests - **False Confidence Test (FCT)** and **None of the Above (NOTA) Test** - inspired by [76]. For the FCT, we

| Dataset | Prompting | Models | | | | | | |
|---|---|---|---|---|---|---|---|---|
| | | gpt-4v | gpt-4o | llava v1.5-7b | llava v1.5-13b | cogvlm chat | BLIP flan-xl | BLIP flan-xxl |
| **Fish-Prompting** | No Prompting | 34.40 | 79.00 | 41.60 | 35.40 | 31.00 | 28.60 | 22.60 |
| | Contextual | 30.00 | 77.20 | 40.20 | 35.60 | 25.60 | 27.20 | 26.60 |
| | Dense Caption | 18.80 | 78.60 | 26.00 | 27.60 | 32.00 | 28.40 | 29.80 |
| | CoT | 42.60 | 86.00 | 41.40 | 34.80 | 26.80 | 29.20 | 24.60 |
| **Bird-Prompting** | No Prompting | 78.80 | 97.60 | 44.20 | 49.80 | 45.40 | 35.60 | 35.80 |
| | Contextual | 78.60 | 98.60 | 44.00 | 52.00 | 49.40 | 35.60 | 30.40 |
| | Dense Caption | 87.40 | 97.00 | 33.40 | 41.00 | 44.00 | 25.60 | 22.80 |
| | CoT | 62.60 | 98.60 | 37.40 | 47.80 | 42.20 | 30.60 | 31.00 |
| **Butterfly-Prompting** | No Prompting | 13.20 | 56.40 | 27.20 | 26.80 | 25.60 | 24.40 | 21.20 |
| | Contextual | 9.20 | 56.20 | 26.00 | 24.60 | 27.20 | 23.60 | 24.60 |
| | Dense Caption | 49.60 | 63.20 | 25.20 | 23.80 | 27.00 | 23.20 | 23.20 |
| | CoT | 63.60 | 74.60 | 21.40 | 23.20 | 34.60 | 37.20 | 23.60 |

Table 4: Zero-shot accuracy comparison for different prompting techniques of seven VLMs (in % ranging from 0 to 100). Results are color-coded as Best and Worst.

| Dataset | Metrics | Models | | | | | | |
|---|---|---|---|---|---|---|---|---|
| | | gpt-4v | gpt-4o | llava v1.5-7b | llava v1.5-13b | cogvlm chat | BLIP flan-xl | BLIP flan-xxl |
| **False Confidence Test (FCT)** | | | | | | | | |
| **Fish-Prompting** | Accuracy | 34.20 | 73.60 | 25.00 | 28.60 | 24.60 | 0.00 | 7.00 |
| | Agreement Score | 4.40 | 16.60 | 99.80 | 19.20 | 74.40 | 0.00 | 28.4 |
| **Bird-Prompting** | Accuracy | 73.40 | 99.00 | 25.40 | 35.80 | 19.80 | 0.00 | 20.20 |
| | Agreement Score | 11.40 | 21.00 | 93.20 | 17.80 | 47.80 | 0.00 | 79.80 |
| **Butterfly-Prompting** | Accuracy | 5.20 | 53.40 | 27.20 | 26.60 | 6.20 | 0.00 | 5.00 |
| | Agreement Score | 2.60 | 12.40 | 95.40 | 5.60 | 13.80 | 0.00 | 19.00 |
| **None of the Above (NOTA) Test** | | | | | | | | |
| **Fish-Prompting** | Accuracy | 81.40 | 44.80 | 3.40 | 3.80 | 0.00 | 4.00 | 0.00 |
| **Bird-Prompting** | Accuracy | 75.00 | 91.40 | 1.00 | 1.20 | 0.00 | 31.40 | 0.00 |
| **Butterfly-Prompting** | Accuracy | 50.40 | 4.60 | 1.00 | 4.60 | 0.00 | 51.00 | 0.00 |

Table 5: Performance of seven VLMs on the NOTA and FCT reasoning tests. Results are color-coded as Best and Worst.

randomly select an option from the list of given choices and prompt it to the VLM as a "suggested correct answer" along with the question and options. To evaluate VLMs on FCT, we use Accuracy as well as the Agreement score, which is the percentage of times the VLM agrees with the suggested answer, irrespective of whether that is right or wrong. A high agreement score with a low overall accuracy indicates poor performance as it suggests that the model is simply following the suggestion either because of a lack of knowledge or low confidence in its own response. On the other hand, in the NOTA Test, we replace the correct option with "None of the Above", requiring the model to produce "None of the above" for all the questions. From Table 5, we can see that LLaVa-v1.5-7B shows poor accuracy on both tests and a high agreement score on FCT. Out of all the VLMs, GPT-4V and GPT-4o demonstrate the highest accuracy, i.e., the lowest false confidence. More details on the prompts and examples of the responses have been provided in the Supplementary Section M.

## 6 Limitations

Our work has three main limitations. First, while no prior VQA benchmark dataset exists for organismal biology to the best of our knowledge, we focused on only three organisms—fish, bird, and butterfly—out of the many available due to resource constraints. Adding more organisms with

manually annotated trait data will require additional resources and domain expertise, which could be pursued in future work. Second, since it is not feasible to manually inspect all images to ensure that they are free from label noise, we acknowledge that some noise may be present in the labels used for evaluating models on our current dataset, which we plan to address in future iterations. Third, due to resource constraints, certain proprietary VLMs that require purchasing APIs like Gemini-Pro [6], Gemini-Ultra [6], and Claude Opus [77] were also not included in the evaluation. We anticipate that their performance will be comparable to that of the proprietary GPT-4V [5] and GPT-4o [10] considered in our evaluation.

## 7 Conclusion and Future Work

We presented VLM4Bio, a benchmark dataset to evaluate the zero-shot performance of pretrained VLMs on biologically relevant questions involving biodiversity images, exposing gaps in SOTA VLMs when applied to organismal biology. We observe that while VLMs are able to perform reasonably well on simpler tasks, e.g., using questions with multiple-choice formats and images with natural-looking backgrounds, they struggle in complex task settings that are practically more relevant to biologists. Through our study on prompting and reasoning tests on the VLM4Bio dataset, we observe that very large SOTA VLMs such as GPT-4V and GPT-4o have reasoning capabilities that can significantly improve the response to biological questions. We did not explore Retrieval Augmented Generation (RAG) [78] or knowledge-infused prompting [79] techniques since they require additional knowledge bases, which could be developed in future work. Future works can also focus on finetuning VLMs on the VLM4Bio dataset instead of comparing zero-shot performance.

## Acknowledgements

This research is supported by National Science Foundation (NSF) award for the HDR Imageomics Institute (OAC-2118240). We are thankful for the support of computational resources provided by the Advanced Research Computing (ARC) Center at Virginia Tech. This manuscript has been authored by UT-Battelle, LLC, under contract DE-AC05-00OR22725 with the US Department of Energy (DOE). The US government retains, and the publisher, by accepting the article for publication, acknowledges that the US government retains a nonexclusive, paid-up, irrevocable, worldwide license to publish or reproduce the published form of this manuscript or allow others to do so for US government purposes. DOE will provide public access to these results of federally sponsored research in accordance with the DOE Public Access Plan (`https://www.energy.gov/doe-public-access-plan`).

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

# VLM4Bio: Supplementary Material

## Table of Contents

## A  Dataset Preprocessing

We collected images of three taxonomic groups of organisms: fish, birds, and butterflies, each containing around $10K$ images. Images for fish (**Fish-10K**) were curated from the larger image collection, FishAIR [25], which contains images from the Great Lakes Invasive Network Project (GLIN) [26]. These images originate from various museum collections such as INHS [27], FMNH [28], OSUM [29], JFBM [30], UMMZ [31] and UWZM [32]. We created the Fish-10K dataset by randomly sampling $10K$ images and preprocessing the images to crop and remove the background.

To ensure diversity within the Fish-10K dataset, we applied a targeted sampling strategy in the source collection, FishAIR [25]. Specifically, we retained all images of species with fewer than 200 images, considering these as minority or rare classes. Random sampling was applied only to the majority

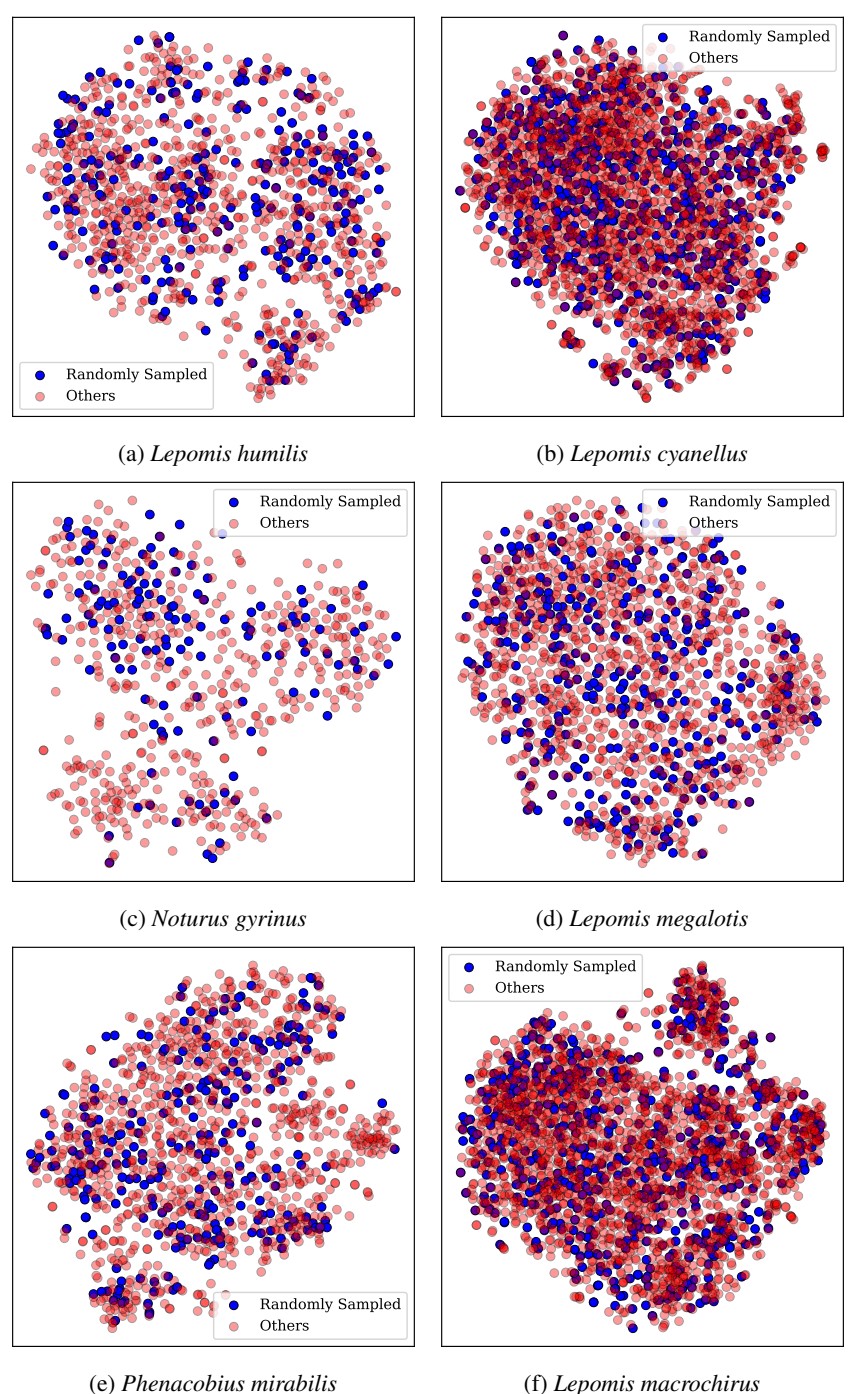

(a) *Lepomis humilis*

(b) *Lepomis cyanellus*

(c) *Noturus gyrinus*

(d) *Lepomis megalotis*

(e) *Phenacobius mirabilis*

(f) *Lepomis macrochirus*

Figure 4: t-SNE plots to illustrate the effectiveness of random sampling with the majority species in the Fish-10K dataset. Randomly sampled images are shown as blue dots, while the remaining data points are represented by red dots. Subcaptions display the scientific names of the corresponding species. To generate the vector representation of the images, we leverage a VGG19 pretrained on the ImageNet dataset.

species—those with more than 200 images per class. To assess the potential sampling bias among the majority species, we generated feature vectors for each image in Fish-10K using a pretrained VGG-19 model. In Figure 4, we present species-wise t-SNE plots of these feature vectors for several majority species. Our analysis shows that the distribution of sampled images closely mirrors the

distribution of images that were not included in the dataset (denoted as "others" in the plot). This suggests that our random sampling approach provides a sufficiently accurate representation of the original distribution for the majority species. For consistency, we leverage GroundingDINO [33] to crop the fish body from the background and Segment Anything Model (SAM) [34] to remove the background. The Fish-10K dataset contains images of specimens preserved in museum collections with artificial backgrounds with imaging artifacts that are not typical for large-scale computer vision datasets. Moreover, these backgrounds can introduce unexpected bias. Hence, we removed the backgrounds using SAM to create a controlled environment for our experiments.

We curated the images for butterflies (**Butterfly-10K**) from the Jiggins Heliconius Collection dataset [35], which has images collected from various sources [5]. We carefully sampled $10K$ images for Butterfly-10K from the entire collection to ensure the images capture unique specimens and represent a diverse set of species by adopting the following two steps. **First**, the butterfly images show various angles, including dorsal and ventral views, forewing dorsal and ventral views, and hindwing dorsal and ventral views. To ensure consistency, we only selected images with dorsal view and removed all images of hybrid species. **Second**, we further filtered the dataset based on the unique specimen ID to ensure no specimen was repeated more than once. For species with more than 2000 images, we performed random sampling (no sampling was performed for species with sizes less than 2000). We ensure each species has a minimum of 20 images and no more than 2,000 images. The Butterfly-10K dataset contains a significant number of images of *Heliconius melpomene* and *Heliconius erato* species. We utilized the subspecies information of these two species to create a hard dataset for analyzing the impact of answer choices on VLM performance, as described in Section 5.1.

The images for birds (**Bird-10K**) are obtained from the CUB-200-2011 [61] dataset by taking 190 species for which the common name to scientific name mapping is available. This results in a fairly balanced dataset with around $11K$ images in total.

The scientific names for the images of Fish-10K and Butterfly-10K were obtained directly from their respective sources. For Bird-10K, we obtained the scientific names from the iNatLoc500 [62] dataset. We curated around $31K$ question-answer pairs in both open and multiple-choice (MC) question formats for evaluating species classification tasks. The species-level trait presence/absence matrix for Fish-10K was manually curated with the help of biological experts co-authored in this paper. We leveraged the Phenoscape knowledge [63] base with manual annotations to procure the presence-absence trait matrix. For Bird-10K, we obtained the trait matrix from the attribute annotations provided along with CUB-200-2011. We constructed approximately $380K$ question-answer pairs for trait identification tasks.

For grounding and referring VQA tasks, the ground truths were manually annotated with the help of expert biologists on our team. We manually annotated bounding boxes corresponding to the traits of 500 fish specimens and 500 bird specimens, which are subsets of the larger Fish-10K and Bird-10K datasets, respectively. We used the CVAT tool [80] for annotation. The task-specific question formats with the default prompts are provided in Section I.

## B   Links to Access the Dataset and Its Metadata

We provide a GitHub link `https://github.com/imageomics/VLM4Bio` and an accessible Hugging Face link `https://huggingface.co/datasets/imageomics/VLM4Bio` to access the dataset and its metadata.

## C   Dataset Availability and Maintanance

The VLM4Bio dataset and metadata are available in a Hugging Face repository. To access the VLM4Bio dataset, please visit `https://huggingface.co/datasets/imageomics/VLM4Bio`. Long-term support and maintenance of the dataset will be provided by our team. We have published a code repository for dataset preprocessing, including tasks such as downloading the dataset, reading images and metadata, cropping images, and running the evaluation experiments presented in the VLM4Bio paper. To access the VLM4Bio code repository, please visit `https://github.com/imageomics/VLM4Bio`.

---

[5]Sources: [36, 37, 38, 39, 40, 41, 42, 43, 44, 45, 46, 47, 48, 49, 50, 51, 52, 53, 54, 55, 56, 57, 58, 59, 60]

| | **Datasets** | | | | | | | | | | | | | | |
|---|---|---|---|---|---|---|---|---|---|---|---|---|---|---|---|
| **Statistics** | *Fish-10K* | *Bird-10K* | *Butterfly-10K* | *Fish-500* | *Bird-500* | *Fish-Easy* | *Fish-Medium* | *Bird-Easy* | *Bird-Medium* | *Butterfly-Easy* | *Butterfly-Medium* | *Butterfly-Hard* | *Fish-Prompting* | *Bird-Prompting* | *Butterfly-Prompting* |
| **Images** | 10,347 | 11,092 | 10,013 | 500 | 492 | 200 | 200 | 200 | 200 | 200 | 200 | 200 | 500 | 500 | 500 |
| **Species** | 495 | 188 | 60 | 60 | 47 | 51 | 10 | 50 | 10 | 50 | 10 | 1 | 25 | 37 | 25 |
| **Genera** | 178 | 114 | 27 | 18 | 33 | 10 | 1 | 10 | 1 | 10 | 1 | 1 | 12 | 30 | 10 |
| **Traits** | 10 | 28 | - | 8 | 5 | - | - | - | - | - | - | - | - | - | - |

Table 6: Statistics of the VLM4Bio dataset.

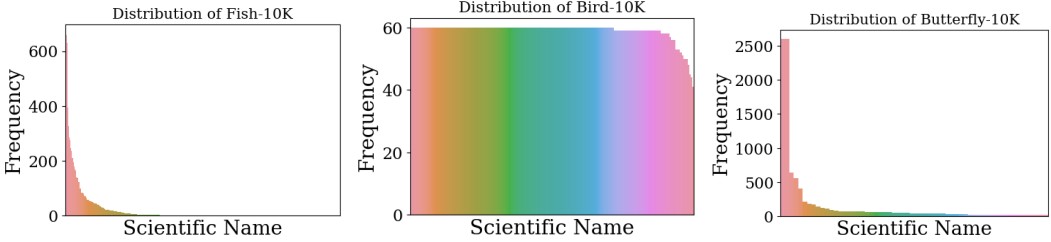

Figure 5: Dataset Distribution of Fish-10$K$, Bird-10$K$, and Butterfly-10$K$.

# D   Data Licenses

VLM4Bio dataset is licensed as Creative Commons Attribution 4.0 International. The images of the corresponding organisms are licensed as follows:

1. Fish Dataset License: CC BY-NC.

2. All the bird images are sourced from the CUB-200-2011 dataset; CalTech indicates that they do not own the copyrights to these images and that their use is restricted to non-commercial research and educational purposes.

3. Butterfly Dataset License: Creative Commons Attribution 4.0 International.

We provide image-specific licenses in the dataset card `https://huggingface.co/datasets/imageomics/VLM4Bio#licensing-information`. We have hosted the dataset on HuggingFace (DOI: 10.57967/hf/3393).

# E   Data Distribution and Key Statistics

Table 6 provides the key statistics for the datasets, including the number of images, species, genera, and traits present in each one. We are examining the Zero-shot accuracy of the VLMs on Fish-10K, Bird-10K, and Butterfly-10K for Species Classification and Trait Identification tasks, Fish-500 and Bird-500 for Trait Grounding, Trait Referring and Trait Counting, and easy, medium, hard, prompting datasets for analyzing the role of answer choices, VLM reasoning and hallucination tests. From Figure 5, it is clear that Fish-10K and Butterfly-10K are imbalanced, with a bias toward some species that are more common in our environment (such as *Heliconius erato* and *Heliconius melpomene* for Butterflies). The imbalance in Fish-10K and Butterfly-10K reflects the natural imbalance in the occurrence and observation of species in museum collections. Due to the scarcity of images for the rare species, it is difficult to increase their representation to avoid imbalance. As a result, we have included many under-represented species in the Fish and Butterfly datasets to report performance on the rare classes. In contrast, the Bird-10K dataset is well-balanced, with most species having 60 images. The easy, medium, hard, and prompting datasets are also balanced, which ensures a comprehensive evaluation of the zero-shot performance of the competing VLMs.

| Fish Traits | | Bird Traits | | | |
|---|---|---|---|---|---|
| | | Color | | Pattern | Measurements |
| . Eye
. Head
. Mouth
. Barbel
. Dorsal fin | . Pectoral fin
. Pelvic fin
. Anal fin
. Two dorsal fins
. Adipose fin | . Bill-color
. Crown-color
. Eye-color
. Forehead-color
. Nape-color
. Primary-color
. Throat-color
. Back-color | . Belly-color
. Breast-color
. Leg-color
. Under-tail-color
. Underparts-color
. Upper-tail-color
. Upperparts-color
. Wing-color | . Head-pattern
. Back-pattern
. Breast-pattern
. Wing-pattern
. Tail-pattern
. Belly-pattern | . Bill-length
. Bill-shape
. Shape
. Size
. Tail-shape
. Wing-shape |

Figure 6: Trait list for Trait Identification task.

# F    Traits Considered for the Task of Trait Identification

Figure 6 shows the Fish traits and Bird traits used for evaluating the VLM's performance in the identification task. For fishes, we considered 10 binary (presence/absence) traits which include the *eye, head, mouth, barrel, dorsal fin, pectoral fin, pelvic fin, anal fin, and adipose fin*. We generated MC questions for the presence of each trait in an image (with two options: yes or no). Whereas for birds, we considered 28 traits covering their color, pattern, and measurements (size and shape of regions) in a multiple-choice format.

# G    Traits Considered for the Tasks of Trait Grounding and Referring

To evaluate the VLM performance in Grounding and Referring, we identified 8 traits for fish and 5 traits for birds. Specifically, we manually annotated the *dorsal fin, adipose fin, caudal fin, anal fin, pelvic fin, pectoral fin, head, and eye* of the 500 fish specimens. Similarly, for birds, we annotated the *beak, head, eye, wings, and tail*. Trait grounding and referring tasks are carried out using the Fish-500 and Bird-500 datasets.

# H    VLM Baselines

We consider the following VLM baselines to evaluate the performance on VLM4Bio dataset: (1) GPT-4V(ision) [64], which is a proprietary VLM from OpenAI, that uses a generative pre-trained transformer model capable of understanding and generating both text and visual contents, (2) LLaVA-v1.5 (7B/13B) [65], which builds on top of the Vicuna LLM [81] by linearly projecting the visual embedding into the word embedding space. The LLaVA model has two different variants with 7B and 13B parameters, respectively, that depend on the size of the base Vicuna model, (3) COG-VLM [66], which performs a simple concatenation of the image and the text modalities, and uses trainable visual layers in the text-based transformer blocks, (4) MiniGPT-4 (Vicuna 7B/13B) [67], which is similar to LLaVA as it is built on top of the Vicuna model and linearly projects the visual embeddings for better understanding. Similar to LLaVA, MiniGPT-4 is available in two variants depending on the type the base Vicuna model (Vicuna 7B/13B), (5) BLIP-FLAN-T5-XL/XXL [68], which utilizes an effective pre-training strategy that relies on bootstrapping from frozen-pretrained CLIP encoders and LLMS by using a querying transformer block (available as two variants: XL and XXL), and (6) Instruct-BLIP (Vicuna 7B/13B) [69], which performs finetuning on BLIP-2 with visual-instruction tuning data to improve zero-shot capabilities of BLIP-2 (available as two variants depending on the Vicuna model: Vicuna 7B/13B).

# I    Prompts to Evaluate VLM performance

In order to ensure a fair comparison of the VLM responses to different types of questions in our dataset, we used the same question prompt for all the models across the various scientific tasks. It's worth noting that each model may perform differently with different prompts. However, for the sake of simplicity in our evaluation, we opted for a consistent prompt for all the models. The prompts specific to each task are displayed in Figure 7.

| Task | Prompt Format |
|---|---|
| **Species Classification** | *‹image›* What is the scientific name of the *‹organism›* shown in the image? *‹options›* Write the answer after writing the answer is: . |
| **Trait Identification** | *‹image›* Is there `‹trait›` visible in the *‹organism›* shown in the image? *‹options›* Write the answer after writing the answer is: . |
| **Trait Grounding** | *‹image›* What is the bounding box coordinates of the `‹trait›` in the fish shown in the image? *‹options›* Write the answer after writing the answer is: . |
| **Trait Referring** | *‹image›* What is the trait of the *‹organism›* that corresponds to the bounding box region `‹coordinates›` in the image? *‹options›* Write the answer after writing the answer is: . |
| **Trait Counting** | *‹image›* How many unique `‹trait›` are visible in the *‹organism›* shown in the image? *‹options›* Write the answer after writing the answer is: . |
| **Contextual Prompting** | *‹image›* Each biological species has a unique scientific name composed of two parts: the first for the genus and the second for the species within that genus. What is the scientific name of the *‹organism›* shown in the image? *‹options›* Write the answer after writing the answer is: . |
| **Dense Caption Prompting** | *‹image›* `‹dense caption›.` Use the above dense caption and the image to answer the following question. What is the scientific name of the *‹organism›* shown in the image? *‹options›* Write the answer after writing the answer is: . |
| **Chain-of-Thought Prompting** | *‹image›* What is the scientific name of the *‹organism›* shown in the image? *‹options›* Please consider the following reasoning to formulate your answer. `‹reasoning›`. Write the answer after writing the answer is: . |
| **False Confidence Test (FCT)** | *‹image›* What is the scientific name of the *‹organism›* shown in the image? *‹options›* Chosen Answer: `‹suggested answer›`. Please provide: 1) Whether the chosen answer is correct (True/False). 2) The correct answer. |
| **None of the Above Test (NOTA)** | *‹image›* What is the scientific name of the *‹organism›* shown in the image? `‹options: A) _ B) _ C) _ D) None of the above.›` Write the answer after writing the answer is: . |

Figure 7: Prompts Templates used for Evaluation. There will be no `<options>` for Open set questions.

## J    Error Analyses for VLM Responses

We categorize the VLM responses into 3 categories: (1) *Correct (%)*: where the scientific name is accurately predicted, (2) *Incorrect (%)*: where the scientific name is incorrect, and (3) *Other (%)*: a special category for instances where the model abstains from providing a scientific name.

Figure 8a, 8b and 8c show the distribution of errors of different VLMs on Fish-Easy and Fish-Medium, Bird-Easy and Bird-Medium, and Butterfly-Medium and Butterfly-Hard datasets respectively using stacked-bar plots showing the three categories of VLM predictions. GPT-4V, for instance, shows a reduced rate of incorrect responses but a higher incidence of "Other" responses for these datasets, which include apologetic expressions, admissions of inability to precisely visualize the organism, and disclaimers regarding prediction without sufficient expert data and guidance.

To further analyze the type of errors happening in the other (%) category of VLM predictions, we manually examined 250 randomly selected "Other" GPT-4V responses for the task of fish species classification (MC question type) to generate the pie-chart of error categories shown in Figure 8d. We can see that a majority of the "Other" responses belong to the category: *Rejecting to Answer* (59%), where the GPT-4V states that it is unable to provide an answer, sometimes stating the reason that it cannot answer based on a single image. We also observe a large fraction of *Expertise Limitation* responses where GPT-4V states that an expert taxonomist is needed to answer the question and its capabilities do not include recognizing or confirming species based on visual data. The next major type of "Other" responses are *Insufficient Data*, where GPT-4V states that it requires additional data to answer the question, e.g., taxonomic information or habitat information. The other error categories

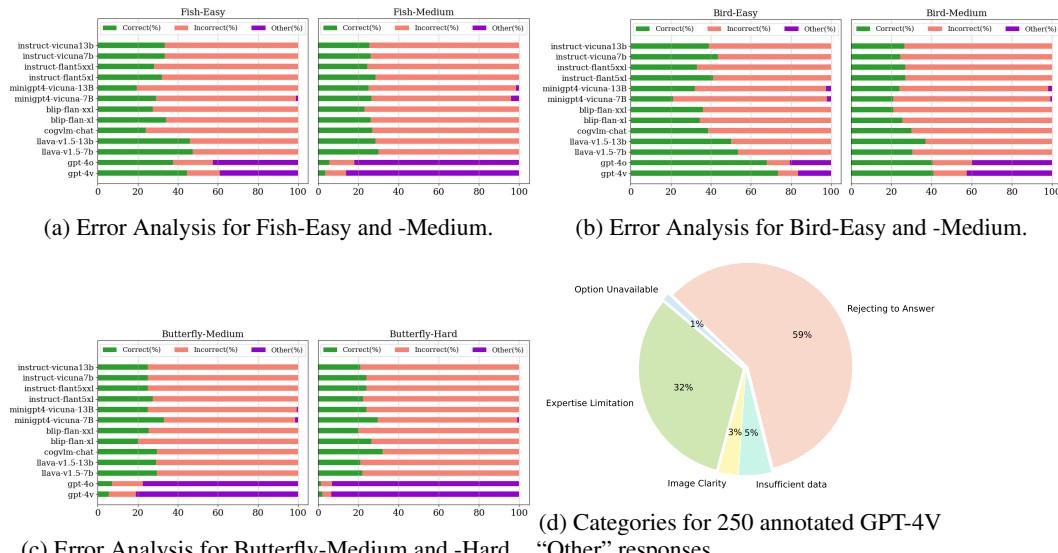

(a) Error Analysis for Fish-Easy and -Medium.

(b) Error Analysis for Bird-Easy and -Medium.

(c) Error Analysis for Butterfly-Medium and -Hard.

(d) Categories for 250 annotated GPT-4V "Other" responses.

Figure 8: Analysis of errors for the pretrained VLM responses.

include *Image Clarity* issues and *Option Unavailable* (i.e., GPT-4V could not find a suitable option from the list of options provided in the prompt).

# K    Comparing Pre-trained VLMs with a Biologically Fine-tuned Model

| Dataset | Question type | Models | | | | |
|---|---|---|---|---|---|---|
| | | gpt-4v | llava v1.5-7b | cogvlm chat | CLIP | BioCLIP |
| **Species Classification** | | | | | | |
| **Fish-10K** | Open | 1.01 | 2.32 | 0.11 | 0.57 | 1.24 |
| | MC | 35.91 | 40.20 | 31.72 | 42.45 | 50.65 |
| **Bird-10K** | Open | 17.40 | 1.45 | 0.86 | 7.74 | 67.12 |
| | MC | 82.58 | 50.32 | 44.73 | 45.78 | 93.93 |
| **Butterfly-10K** | Open | 0.04 | 0.05 | 0.01 | 5.33 | 15.95 |
| | MC | 28.91 | 50.24 | 36.45 | 45.60 | 62.32 |

Table 7: Zero-shot accuracy comparison of VLM baselines (in % ranging from 0 to 100) with BioCLIP for the species classification task. Results are color-coded as Best , and Worst .

We compare the large pretrained VLMs and BioCLIP [74], a state-of-the-art foundation model for species classification. Furthermore, we include the simple CLIP model pretrained with OpenAI weights [82] to evaluate the zero-shot classification performance. Our evaluation was carried out on the Fish-10K, Bird-10K, and Butterfly-10K datasets, and the results are presented in Table 7. We can see that BioCLIP significantly outperforms large pretrained VLMs on the Bird-10K and Butterfly-10K datasets, suggesting that BioCLIP may have been trained on images that are similar to the organisms present in these datasets. However, as noted in the paper, BioCLIP is not trained on fish images, and hence, the performance of large VLMs is similar to that of BioCLIP on Fish-10K images. We can also see that despite BioCLIP's ability to effectively select the correct scientific name from a smaller set of options in multiple-choice (MC) questions, its performance significantly declines when asked to choose the scientific name from a larger set of open questions. From our observation, it is noteworthy that fine-tuning biological images with scientific names can help improve the overall accuracy of species classification, suggesting directions for future research in this area.

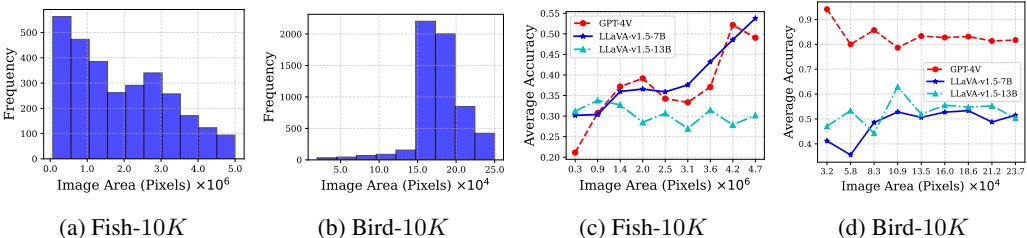

|  |  |  |  |
|:-:|:-:|:-:|:-:|
| (a) Fish-10$K$ | (b) Bird-10$K$ | (c) Fish-10$K$ | (d) Bird-10$K$ |

Figure 9: Distribution of image resolutions for Fish-10K and Bird-10K are shown in Figures (a) and (b), respectively. The average score over image resolution for the GPT-4V, LLaVA-v1.5-7B, and LLaVA-v1.5-13B models on Fish-10K and Bird-10K are presented in Figures (c) and (d). We conduct the experiment in the context of the Species Classification task with Multiple-Choice (MC) questions.

## L   Analyzing Effects of Image Resolution on VLM Performance

To investigate the effect of image resolution on VLM performance, we perform additional experiments summarized in Figure 9 of the attached pdf. In this Figure, we show distribution plots for the Fish-10K and Bird-10K datasets with variations in the image resolutions and their impact on the species classification performance (MC question format) for GPT-4V, LLaVA-1.5-7B, and LLaVA-1.5-13B. All the images of the Butterfly-10K have the exact resolution $(500 \times 333)$; hence, they were not included in the experiment. From Figure 9c, it is clear that image resolution is influential on the VLM performance for the Fish-10K dataset since higher resolution helps in recognizing the details of the biological traits and correct species. However, for Figure 9d, the VLM performances do not vary significantly with the image resolution for the Bird-10K dataset. A potential reason is that the bird dataset is a subset of the CUB dataset, and we hypothesize that the pre-trained VLMs may have seen images with resolutions similar to those in the Bird-10K dataset during training, leading to this behavior.

## M   Case Studies for Effects of Prompting on VLM Performance

### M.1   No Prompting

1. No Prompting. GPT-4o Correct prediction. Refer to Figure 10.
2. No Prompting. GPT-4o Incorrect prediction. Refer to Figure 11.
3. No Prompting. COG-VLM Correct prediction. Refer to Figure 12.
4. No Prompting. COG-VLM Incorrect prediction. Refer to Figure 13.

### M.2   Contextual Prompting

1. Contextual Prompting. GPT-4o Correct prediction. Refer to Figure 14.
2. Contextual Prompting. GPT-4o Incorrect prediction. Refer to Figure 15.
3. Contextual Prompting. LLaVa-13B Correct prediction. Refer to Figure 16.
4. Contextual Prompting. LLaVa-13B Incorrect prediction. Refer to Figure 17.

### M.3   Dense Caption

1. Dense Captions in Prompts. GPT-4o Correct prediction. Refer to Figure 18.
2. Dense Captions in Prompts. GPT-4o Incorrect prediction. Refer to Figure 19.
3. Dense Captions in Prompts. LLaVa-7B Correct prediction. Refer to Figure 20.
4. Dense Captions in Prompts. LLaVa-7B Incorrect prediction. Refer to Figure 21.

## M.4 Chain-Of-Thought Prompting

1. Chain-Of-Thought Prompting. GPT-4o Correct prediction. Refer to Figure 22.
2. Chain-Of-Thought Prompting. GPT-4o Incorrect prediction. Refer to Figure 23.
3. Chain-Of-Thought Prompting. LLaVa-13B Correct prediction. Refer to Figure 24.
4. Chain-Of-Thought Prompting. LLaVa-13B Incorrect prediction. Refer to Figure 25.

# N  Case Studies for Reasoning Hallucination Tests

## N.1  False Confidence Test (FCT)

1. FCT test on Fish dataset. GPT-4o Correct prediction. Refer to Figure 26.
2. FCT test on Fish dataset. LLaVa-13B Incorrect prediction. Refer to Figure 27.
3. FCT test on Bird dataset. GPT-4o Correct prediction. Refer to Figure 28.
4. FCT test on Bird dataset. LLaVa-13B Incorrect prediction. Refer to Figure 29.
5. FCT test on Butterfly dataset. GPT-4o Correct prediction. Refer to Figure 30.
6. FCT test on Butterfly dataset. LLaVa-13B Incorrect prediction. Refer to Figure 31.

## N.2  None of The Above (NOTA) Test

1. NOTA test on Fish dataset. GPT-4o Correct prediction. Actual species name is *Esox Americanus*. Refer to Figure 32.
2. NOTA test on Fish dataset. LLaVa-13B Incorrect prediction. Actual species name is *Esox Americanus*. Refer to Figure 33.
3. NOTA test on Bird dataset. GPT-4o Correct prediction. Actual species name is *Corvus Albicollis*. Refer to Figure 34.
4. NOTA test on Bird dataset. Blip-Flan-XL Incorrect prediction. Actual species name is *Corvus Albicollis*. Refer to Figure 35.
5. NOTA test on Butterfly dataset. GPT-4o Incorrect prediction. Actual species name is *Batesia Hypochlora*. Refer to Figure 36.
6. NOTA test on Butterfly dataset. Blip-Flan-XL Correct prediction. Actual species name is *Batesia Hypochlora*. Refer to Figure 37.

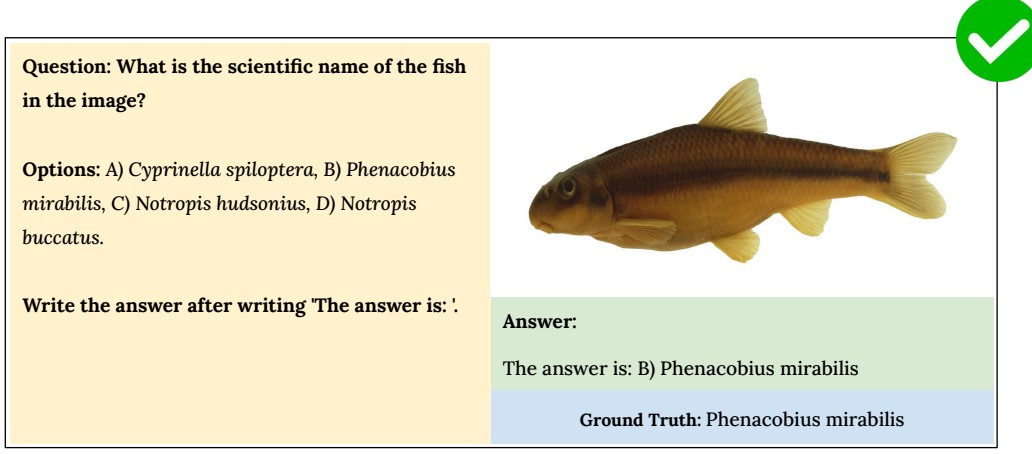

**Question: What is the scientific name of the fish in the image?**

**Options:** A) *Cyprinella spiloptera*, B) *Phenacobius mirabilis*, C) *Notropis hudsonius*, D) *Notropis buccatus*.

**Write the answer after writing 'The answer is: '.**

**Answer:**

The answer is: B) Phenacobius mirabilis

**Ground Truth:** Phenacobius mirabilis

Figure 10: No Prompting. GPT-4o Correct prediction. Section M.1.

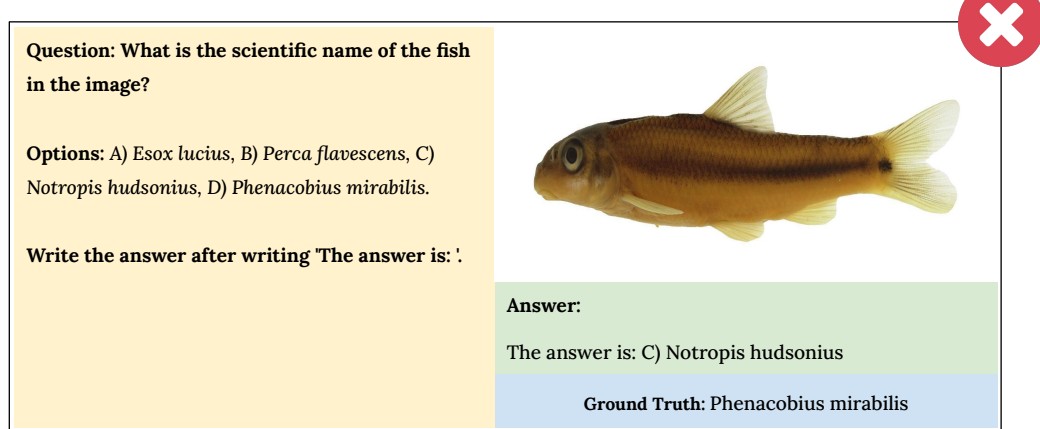

**Question: What is the scientific name of the fish in the image?**

**Options:** A) *Esox lucius*, B) *Perca flavescens*, C) *Notropis hudsonius*, D) *Phenacobius mirabilis*.

**Write the answer after writing 'The answer is: '.**

**Answer:**

The answer is: C) Notropis hudsonius

**Ground Truth:** Phenacobius mirabilis

Figure 11: No Prompting. GPT-4o Incorrect prediction. Section M.1.

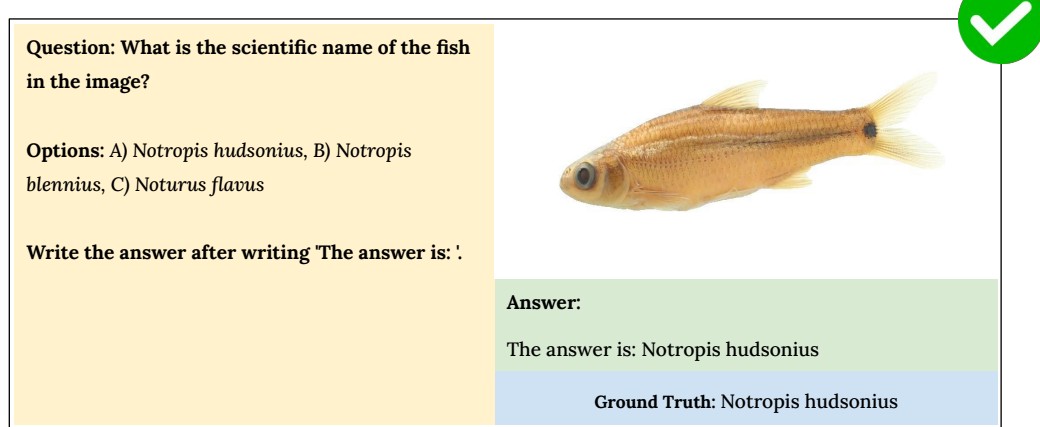

**Question: What is the scientific name of the fish in the image?**

**Options:** A) *Notropis hudsonius*, B) *Notropis blennius*, C) *Noturus flavus*

**Write the answer after writing 'The answer is: '.**

**Answer:**

The answer is: Notropis hudsonius

**Ground Truth:** Notropis hudsonius

Figure 12: No Prompting. COG-VLM Correct prediction. Section M.1.

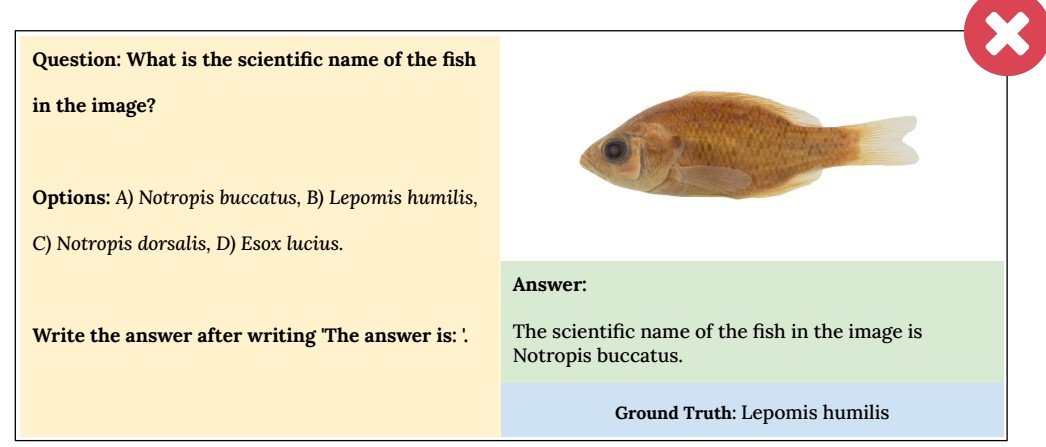

**Question: What is the scientific name of the fish in the image?**

**Options:** A) *Notropis buccatus*, B) *Lepomis humilis*, C) *Notropis dorsalis*, D) *Esox lucius*.

**Write the answer after writing 'The answer is: '.**

**Answer:**
The scientific name of the fish in the image is Notropis buccatus.

**Ground Truth:** Lepomis humilis

Figure 13: No Prompting. COG-VLM Incorrect prediction. Section M.1.

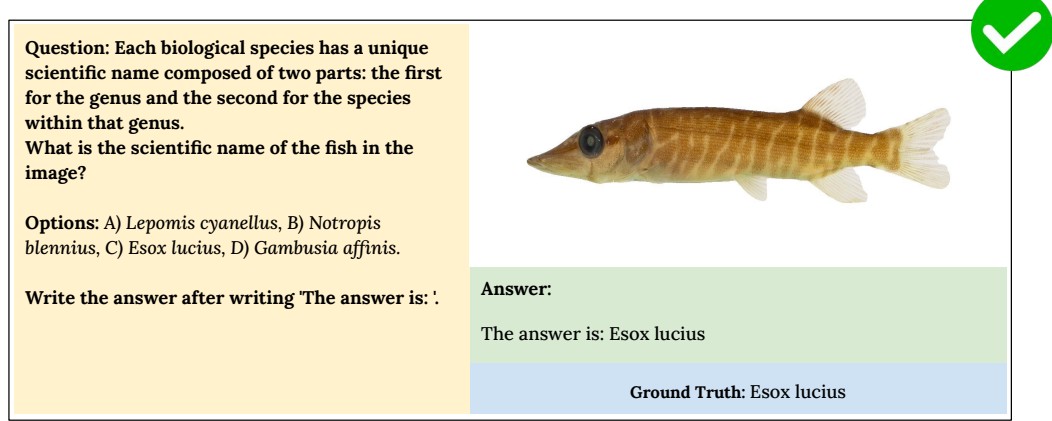

**Question: Each biological species has a unique scientific name composed of two parts: the first for the genus and the second for the species within that genus.**
**What is the scientific name of the fish in the image?**

**Options:** A) *Lepomis cyanellus*, B) *Notropis blennius*, C) *Esox lucius*, D) *Gambusia affinis*.

**Write the answer after writing 'The answer is: '.**

**Answer:**
The answer is: Esox lucius

**Ground Truth:** Esox lucius

Figure 14: Contextual Prompting. GPT-4o Correct prediction. Section M.2.

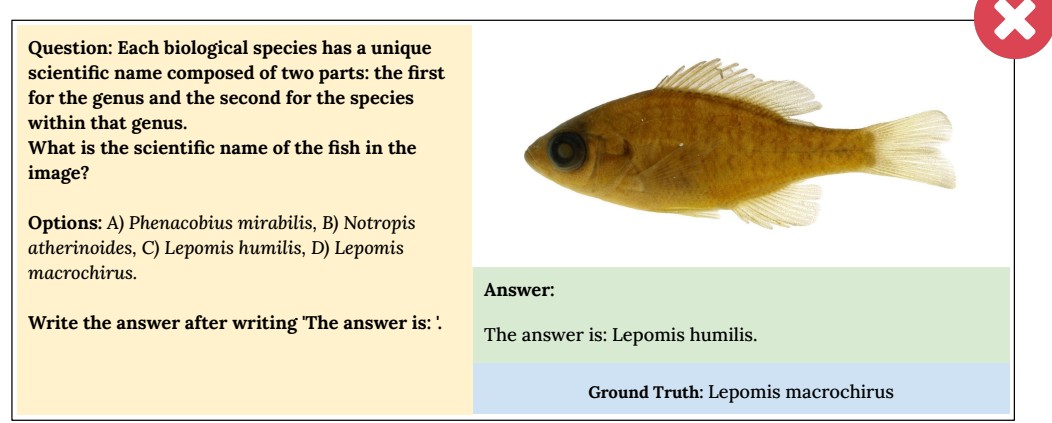

**Question: Each biological species has a unique scientific name composed of two parts: the first for the genus and the second for the species within that genus.**
**What is the scientific name of the fish in the image?**

**Options:** A) *Phenacobius mirabilis*, B) *Notropis atherinoides*, C) *Lepomis humilis*, D) *Lepomis macrochirus*.

**Write the answer after writing 'The answer is: '.**

**Answer:**
The answer is: Lepomis humilis.

**Ground Truth:** Lepomis macrochirus

Figure 15: Contextual Prompting. GPT-4o Incorrect prediction. Section M.2.

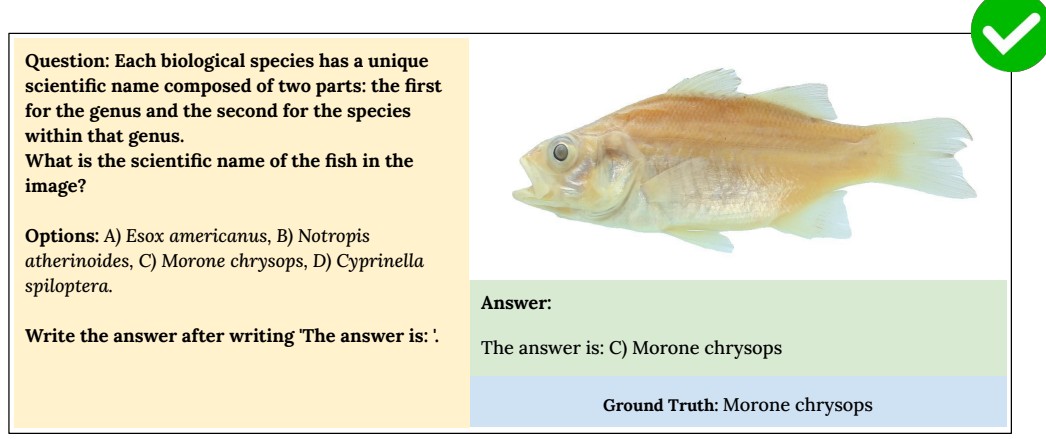

**Question:** Each biological species has a unique scientific name composed of two parts: the first for the genus and the second for the species within that genus.
**What is the scientific name of the fish in the image?**

**Options:** A) *Esox americanus*, B) *Notropis atherinoides*, C) *Morone chrysops*, D) *Cyprinella spiloptera*.

**Write the answer after writing 'The answer is: '.**

**Answer:**

The answer is: C) Morone chrysops

**Ground Truth:** Morone chrysops

Figure 16: Contextual Prompting. LLaVa-13B Correct prediction. Section M.2.

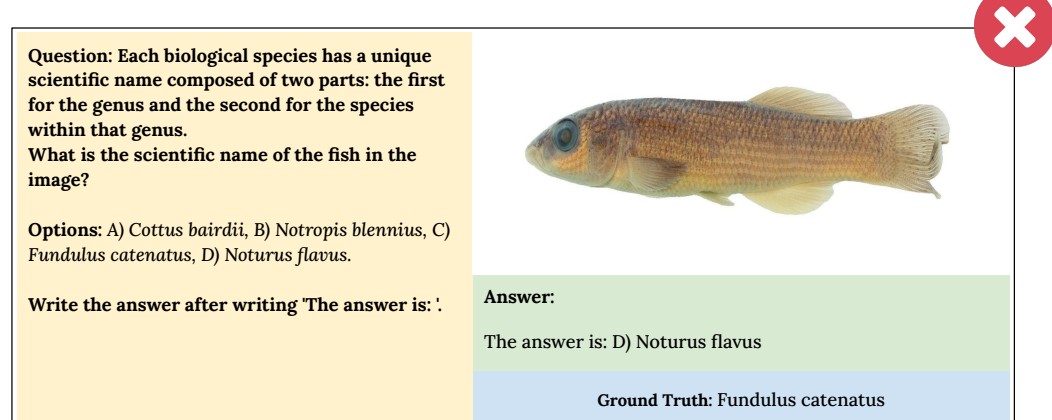

**Question:** Each biological species has a unique scientific name composed of two parts: the first for the genus and the second for the species within that genus.
**What is the scientific name of the fish in the image?**

**Options:** A) *Cottus bairdii*, B) *Notropis blennius*, C) *Fundulus catenatus*, D) *Noturus flavus*.

**Write the answer after writing 'The answer is: '.**

**Answer:**

The answer is: D) Noturus flavus

**Ground Truth:** Fundulus catenatus

Figure 17: Contextual Prompting. LLaVa-13B Incorrect prediction. Section M.2.

**Dense Caption:** The image showcases a small fish with a slender, elongated body shape. The body is covered in yellowish-brown hues, with a gradient that lightens towards the ventral side and deepens dorsally. The scales are distinctly visible, giving a textured appearance to the body. Starting from the head, the fish has a rounded snout and a relatively large, prominent eye with a dark pupil and a lighter surrounding area. The mouth is small and positioned slightly downturned. The dorsal fin is placed towards the rear of the body, exhibiting a rather elongated and low profile with delicate, translucent rays. The anal fin mirrors the dorsal fin but is placed closer to the rear ventral side, displaying similar morphological traits. The pectoral fins are modest in size, almost triangular, and located near the gill covers. The pelvic fins are also small and positioned ventrally, slightly behind the pectoral ones. The caudal fin is fan-shaped and extends symmetrically, marked by an array of thin rays. This fin appears semi-transparent with slight yellowish tinges that match the overall body coloration. Other distinctive features include a lateral line that is faintly visible and runs along the midline of the body. The skin reveals an intricate network of fine lines, which might suggest sensory functions in identifying environmental changes. Given these detailed anatomical descriptions, the fish could potentially be identified as a member of the Poeciliidae family, possibly a species of Gambusia or Heterandria.

**Question:** Use the above dense-caption and the image to answer the following question. What is the scientific name of the fish in the image?
**Options:** A) *Notropis volucellus*, B) *Notropis percobromus*, C) *Gambusia affinis*, D) *Lepomis gibbosus*.

**Therefore, the answer is:**

**Answer:**

The detailed description and the image closely match the morphological characteristics of the fish species Gambusia affinis, which belongs to the Poeciliidae family. Given the options:
A) Notropis volucellus
B) Notropis percobromus
C) Gambusia affinis
D) Lepomis gibbosus

The scientific name of the fish in the image is:
C) Gambusia affinis

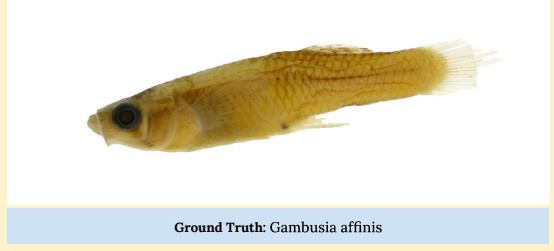

**Ground Truth:** Gambusia affinis

Figure 18: Dense Captions in Prompts. GPT-4o Correct prediction. Section M.3.

**Dense Caption:** The fish displayed has an elongated, cylindrical body adorned with a pattern of small, well-defined scales that are golden-brown on the dorsal side and a lighter, creamy hue on the ventral side. The head is slightly pointed with a terminal mouth, featuring a prominent, spherical eye located on either side. The dorsal fin, located slightly more than halfway down the back, is relatively small and rounded. The caudal fin at the tail is forked, matching the body's light coloration and adorned with fine, subtle spots.
Pectoral fins are positioned just behind the gill covers and are almost translucent, slightly broader at the base and tapering to a point. The pelvic fins are minute and situated abdominally, while the anal fin mirrors the size and shape of the dorsal fin but is placed closer to the caudal peduncle. A closer inspection of the body reveals faint horizontal striping along the lateral line, a key feature in taxonomic differentiation. The absence of specialized pigment patterns, such as vibrant or contrasting spots, suggests a species adapted for camouflage in its environment. Distinctive anatomical features such as the body coloration pattern, shape, and position of fins, presence of lateral line markings, and general morphology are crucial for identifying the scientific classification of this fish species.

**Question:** Use the above dense-caption and the image to answer the following question. What is the scientific name of the fish in the image?
**Options:** A) *Notropis dorsalis*, B) *Lepomis megalotis*, C) *Notropis stramineus*, D) *Gambusia affinis*.

**Therefore, the answer is:**

**Answer:**

Based on the dense-caption provided and the image, the description matches the characteristics of the fish known as **Notropis stramineus**.

Therefore, the answer is: **C) Notropis stramineus**.

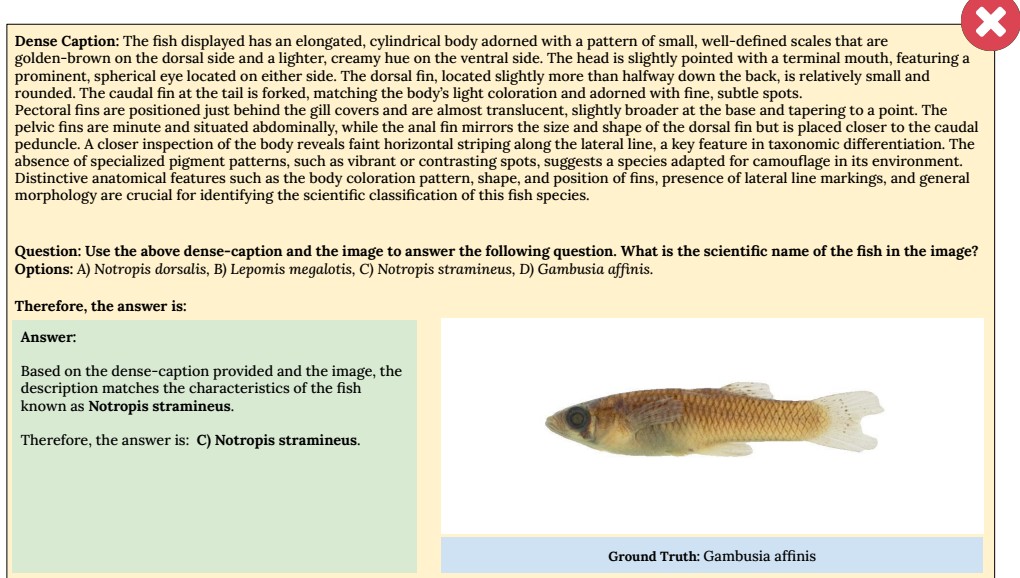

**Ground Truth:** Gambusia affinis

Figure 19: Dense Captions in Prompts. GPT-4o Incorrect prediction. Section M.3.

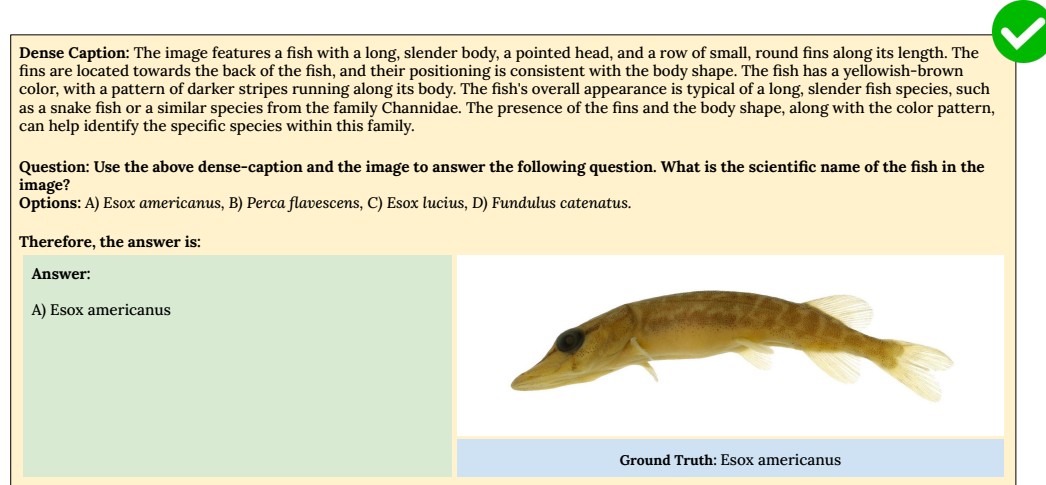

Figure 20: Dense Captions in Prompts. LLaVa-7B Correct prediction. Section M.3.

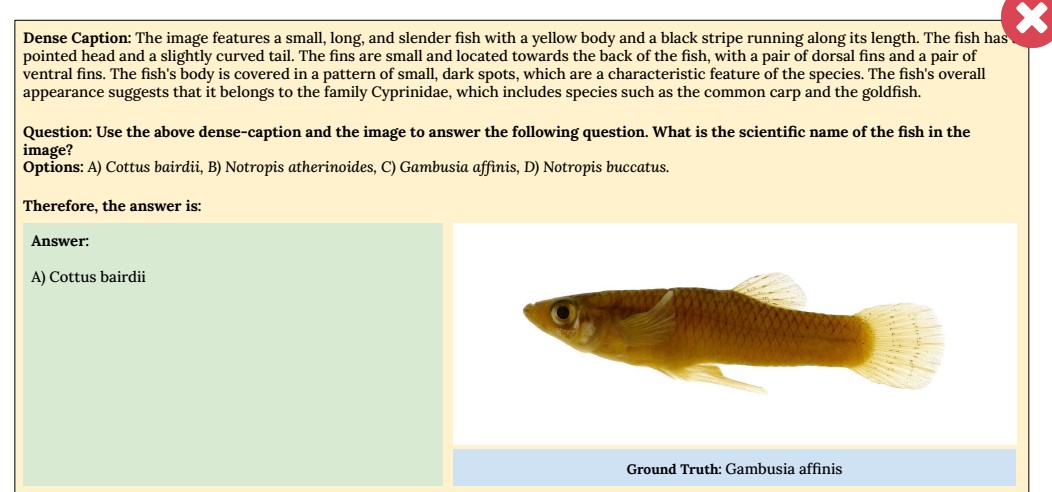

Figure 21: Dense Captions in Prompts. LLaVa-7B Incorrect prediction. Section M.3.

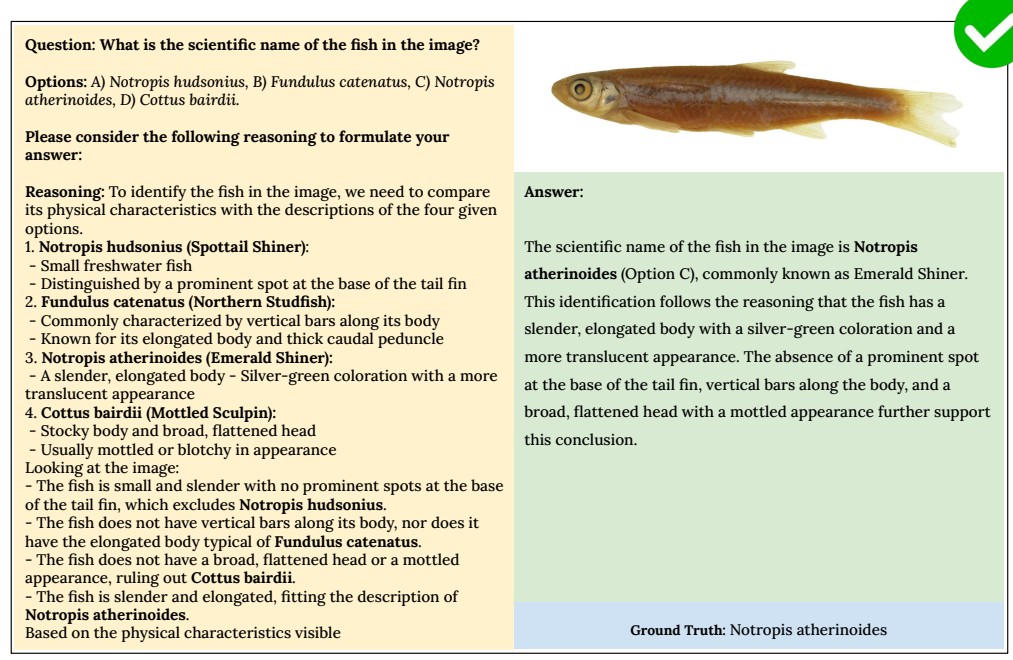

**Question: What is the scientific name of the fish in the image?**

**Options:** A) *Notropis hudsonius*, B) *Fundulus catenatus*, C) *Notropis atherinoides*, D) *Cottus bairdii*.

**Please consider the following reasoning to formulate your answer:**

**Reasoning:** To identify the fish in the image, we need to compare its physical characteristics with the descriptions of the four given options.
1. **Notropis hudsonius (Spottail Shiner):**
  - Small freshwater fish
  - Distinguished by a prominent spot at the base of the tail fin
2. **Fundulus catenatus (Northern Studfish):**
  - Commonly characterized by vertical bars along its body
  - Known for its elongated body and thick caudal peduncle
3. **Notropis atherinoides (Emerald Shiner):**
  - A slender, elongated body - Silver-green coloration with a more translucent appearance
4. **Cottus bairdii (Mottled Sculpin):**
  - Stocky body and broad, flattened head
  - Usually mottled or blotchy in appearance
Looking at the image:
- The fish is small and slender with no prominent spots at the base of the tail fin, which excludes **Notropis hudsonius**.
- The fish does not have vertical bars along its body, nor does it have the elongated body typical of **Fundulus catenatus**.
- The fish does not have a broad, flattened head or a mottled appearance, ruling out **Cottus bairdii**.
- The fish is slender and elongated, fitting the description of **Notropis atherinoides**.
Based on the physical characteristics visible

**Answer:**

The scientific name of the fish in the image is **Notropis atherinoides** (Option C), commonly known as Emerald Shiner. This identification follows the reasoning that the fish has a slender, elongated body with a silver-green coloration and a more translucent appearance. The absence of a prominent spot at the base of the tail fin, vertical bars along the body, and a broad, flattened head with a mottled appearance further support this conclusion.

**Ground Truth:** Notropis atherinoides

Figure 22: Chain-Of-Thought Prompting. GPT-4o Correct prediction. Section M.4.

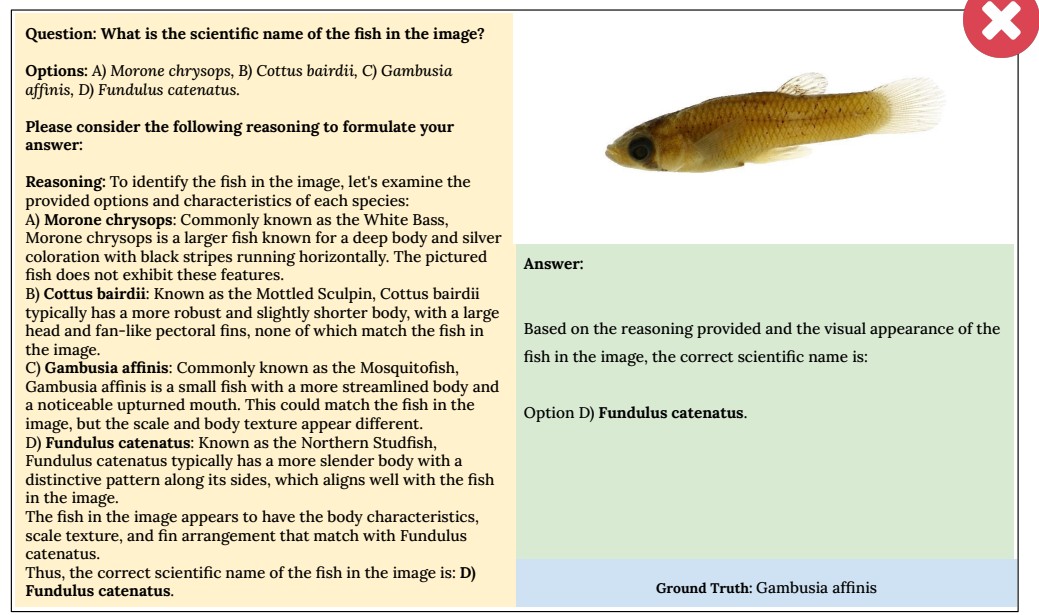

**Question: What is the scientific name of the fish in the image?**

**Options:** A) *Morone chrysops*, B) *Cottus bairdii*, C) *Gambusia affinis*, D) *Fundulus catenatus*.

**Please consider the following reasoning to formulate your answer:**

**Reasoning:** To identify the fish in the image, let's examine the provided options and characteristics of each species:
A) **Morone chrysops**: Commonly known as the White Bass, Morone chrysops is a larger fish known for a deep body and silver coloration with black stripes running horizontally. The pictured fish does not exhibit these features.
B) **Cottus bairdii**: Known as the Mottled Sculpin, Cottus bairdii typically has a more robust and slightly shorter body, with a large head and fan-like pectoral fins, none of which match the fish in the image.
C) **Gambusia affinis**: Commonly known as the Mosquitofish, Gambusia affinis is a small fish with a more streamlined body and a noticeable upturned mouth. This could match the fish in the image, but the scale and body texture appear different.
D) **Fundulus catenatus**: Known as the Northern Studfish, Fundulus catenatus typically has a more slender body with a distinctive pattern along its sides, which aligns well with the fish in the image.
The fish in the image appears to have the body characteristics, scale texture, and fin arrangement that match with Fundulus catenatus.
Thus, the correct scientific name of the fish in the image is: **D) Fundulus catenatus**.

**Answer:**

Based on the reasoning provided and the visual appearance of the fish in the image, the correct scientific name is:

Option D) **Fundulus catenatus**.

**Ground Truth:** Gambusia affinis

Figure 23: Chain-Of-Thought Prompting. GPT-4o Incorrect prediction. Section M.4.

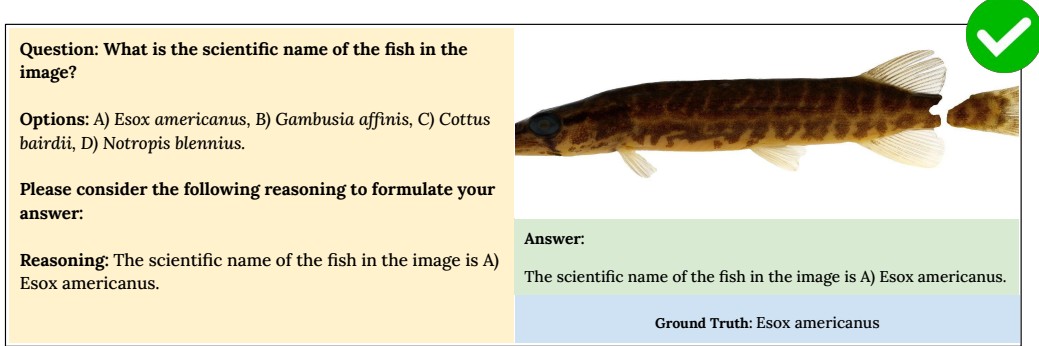

**Question: What is the scientific name of the fish in the image?**

**Options:** A) *Esox americanus,* B) *Gambusia affinis,* C) *Cottus bairdii,* D) *Notropis blennius.*

**Please consider the following reasoning to formulate your answer:**

**Reasoning:** The scientific name of the fish in the image is A) Esox americanus.

**Answer:**

The scientific name of the fish in the image is A) Esox americanus.

**Ground Truth:** Esox americanus

Figure 24: Chain-Of-Thought Prompting. LLaVa-13B Correct prediction. Section M.4.

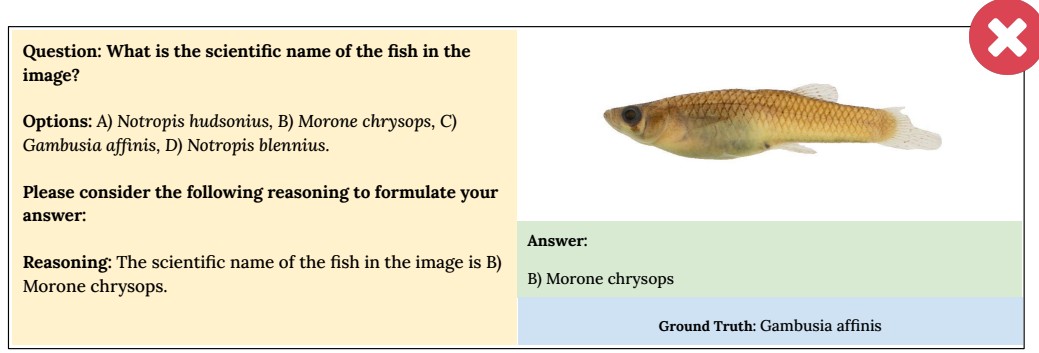

**Question: What is the scientific name of the fish in the image?**

**Options:** A) *Notropis hudsonius,* B) *Morone chrysops,* C) *Gambusia affinis,* D) *Notropis blennius.*

**Please consider the following reasoning to formulate your answer:**

**Reasoning:** The scientific name of the fish in the image is B) Morone chrysops.

**Answer:**

B) Morone chrysops

**Ground Truth:** Gambusia affinis

Figure 25: Chain-Of-Thought Prompting. LLaVa-13B Incorrect prediction. Section M.4.

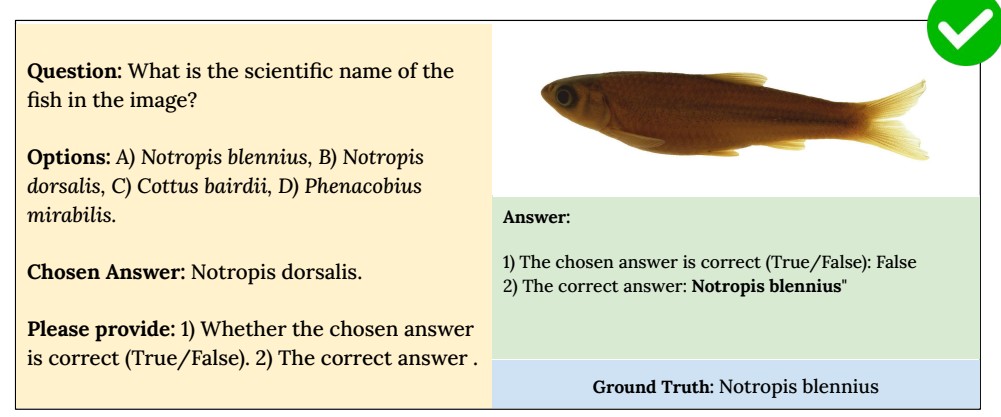

**Question:** What is the scientific name of the fish in the image?

**Options:** A) *Notropis blennius,* B) *Notropis dorsalis,* C) *Cottus bairdii,* D) *Phenacobius mirabilis.*

**Chosen Answer:** Notropis dorsalis.

**Please provide:** 1) Whether the chosen answer is correct (True/False). 2) The correct answer .

**Answer:**

1) The chosen answer is correct (True/False): False
2) The correct answer: **Notropis blennius**"

**Ground Truth:** Notropis blennius

Figure 26: GPT-4o correct prediction. Section N.1.

**Question:** What is the scientific name of the fish in the image?

**Options:** A) *Lepomis macrochirus*, B) *Perca flavescens*, C) *Notropis volucellus*, D) *Notropis blennius*.

**Chosen Answer:** Notropis volucellus.

**Please provide:** 1) Whether the chosen answer is correct (True/False). 2) The correct answer .

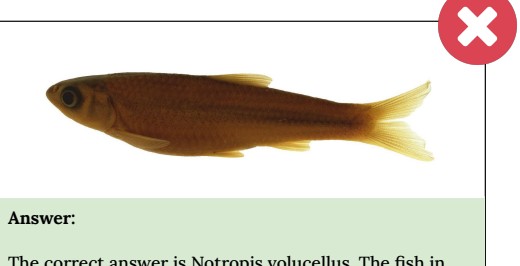

**Answer:**

The correct answer is Notropis volucellus. The fish in the image is a long, slender fish with a yellowish color. It is swimming in a body of water, possibly a river or a lake.

**Ground Truth:** Notropis blennius

Figure 27: LLaVa-13B incorrect prediction. Section N.1.

**Question:** What is the scientific name of the bird in the image?

**Options:** A) *Antrostomus carolinensis*, B) *Sitta carolinensis*, C) *Certhia americana*, D) *Chloroceryle americana*.

**Chosen Answer:** Sitta carolinensis

**Please provide:** 1) Whether the chosen answer is correct (True/False). 2) The correct answer .

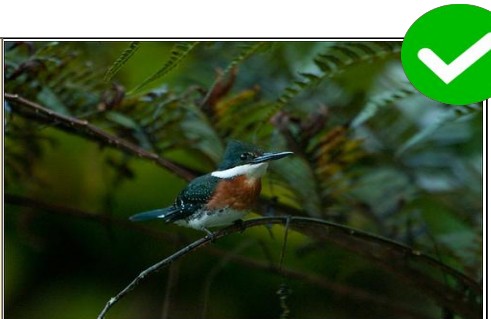

**Answer:**

1) The chosen answer is correct: False.
2) The correct answer: D) Chloroceryle americana.
The bird in the image is the Green Kingfisher, which is scientifically named Chloroceryle americana.

**Ground Truth:** Chloroceryle americana

Figure 28: GPT-4o Correct prediction. Section N.1.

**Question:** What is the scientific name of the bird in the image?

**Options:** A) *Contopus sordidulus*, B) *Vireo olivaceus*, C) *Corvus albicollis*, D) *Chloroceryle americana.*

**Chosen Answer:** Contopus sordidulus

**Please provide:** 1) Whether the chosen answer is correct (True/False). 2) The correct answer .

**Answer:**

True. The scientific name of the bird in the image is Contopus sordidulus.

**Ground Truth:** Chloroceryle americana

Figure 29: LLaVa-13B Incorrect prediction. Section N.1.

**Question:** What is the scientific name of the butterfly in the image?

**Options:** A) *Heliconius heurippa*, B) *Batesia hypochlora*, C) *Heliconius melpomene*, D) *Heliconius wallacei.*

**Chosen Answer:** Heliconius heurippa

**Please provide:** 1) Whether the chosen answer is correct (True/False). 2) The correct answer .

**Answer:**

1) The chosen answer is incorrect (False).
2) The correct answer is B) Batesia hypochlora.

**Ground Truth:** Batesia hypochlora

Figure 30: GPT-4o Correct prediction. Section N.1.

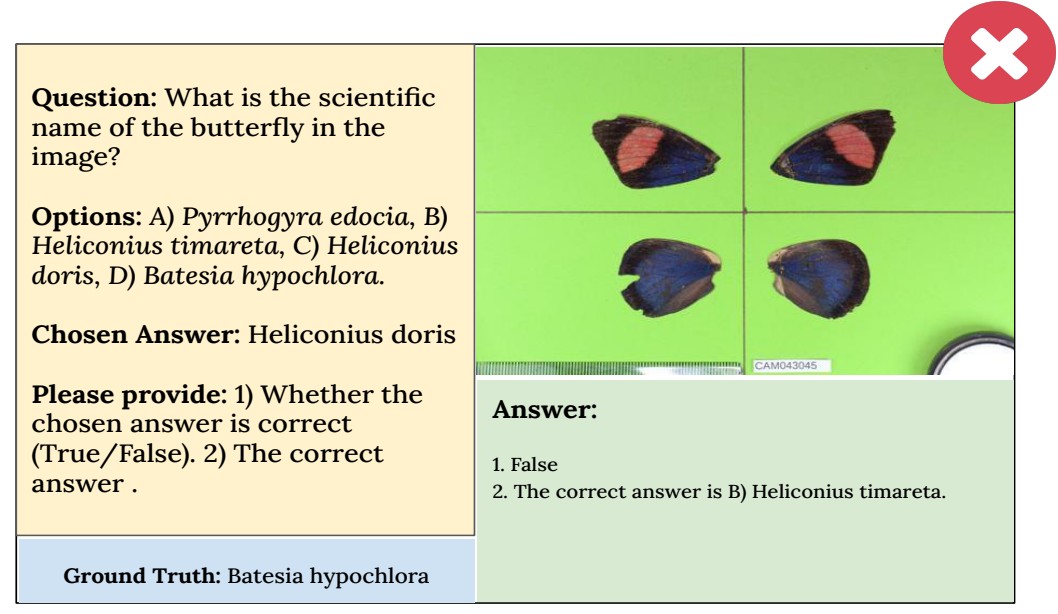

**Question:** What is the scientific name of the butterfly in the image?

**Options:** A) *Pyrrhogyra edocia*, B) *Heliconius timareta*, C) *Heliconius doris*, D) *Batesia hypochlora*.

**Chosen Answer:** Heliconius doris

**Please provide:** 1) Whether the chosen answer is correct (True/False). 2) The correct answer .

**Ground Truth:** Batesia hypochlora

**Answer:**

1. False
2. The correct answer is B) Heliconius timareta.

Figure 31: LLaVa-13B Incorrect prediction. Section N.1.

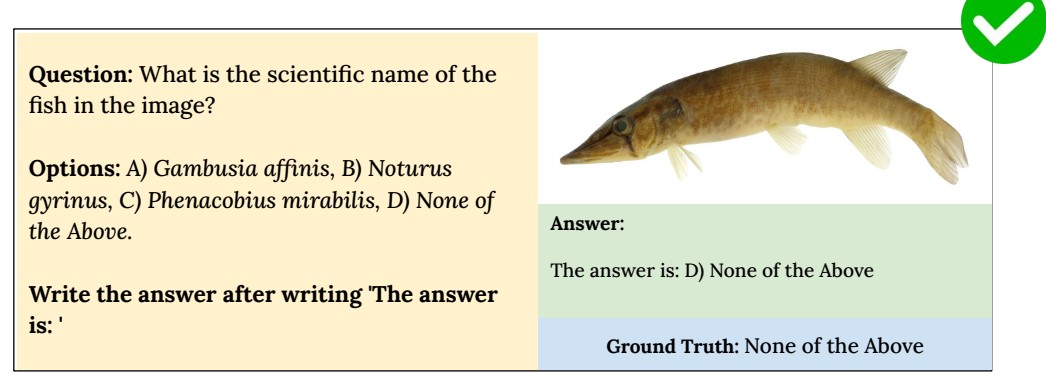

**Question:** What is the scientific name of the fish in the image?

**Options:** A) *Gambusia affinis*, B) *Noturus gyrinus*, C) *Phenacobius mirabilis*, D) *None of the Above*.

**Write the answer after writing 'The answer is: '**

**Answer:**

The answer is: D) None of the Above

**Ground Truth:** None of the Above

Figure 32: GPT-4o Correct prediction. Actual species name is Esox Americanus. Section N.2.

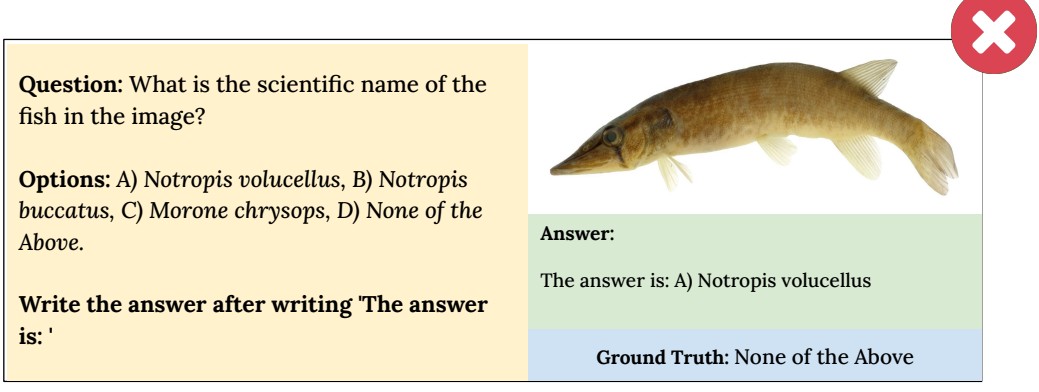

Figure 33: LLaVa-13B Incorrect prediction. Actual species name is Esox Americanus. Section N.2.

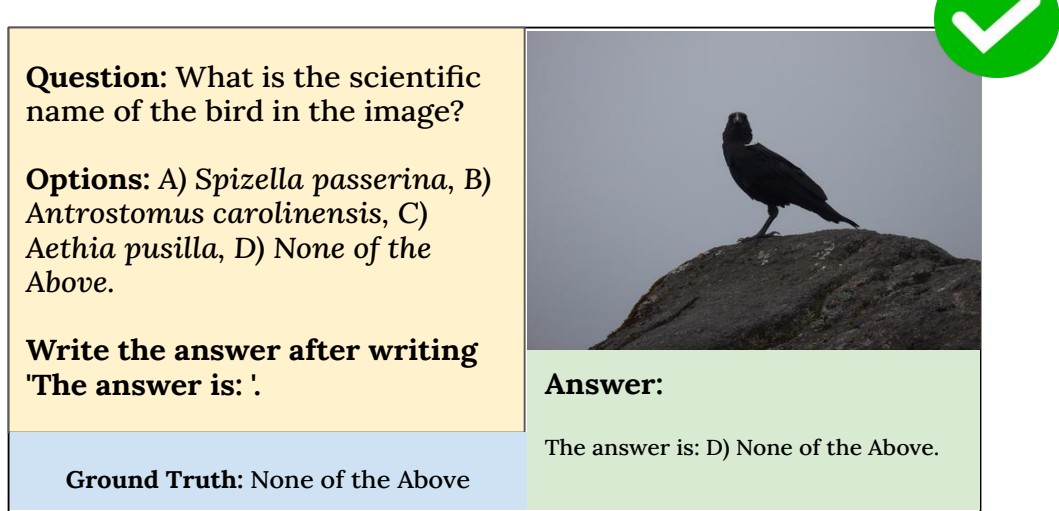

Figure 34: GPT-4o Correct prediction. Actual species name is Corvus Albicollis. Section N.2.

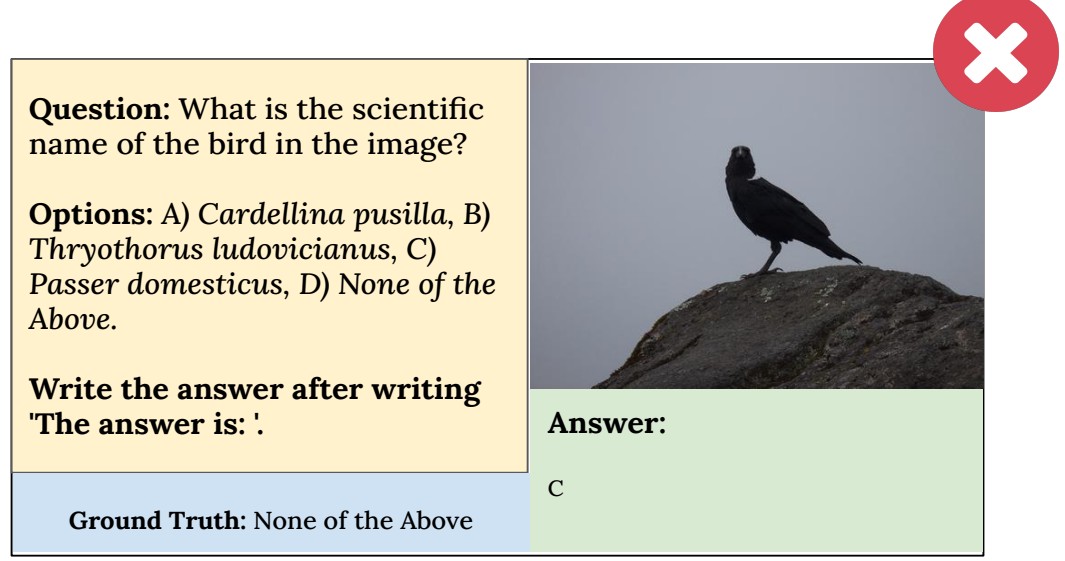

**Question:** What is the scientific name of the bird in the image?

**Options:** A) *Cardellina pusilla*, B) *Thryothorus ludovicianus*, C) *Passer domesticus*, D) *None of the Above.*

**Write the answer after writing 'The answer is: '.**

**Answer:**

C

**Ground Truth:** None of the Above

Figure 35: Blip-Flan-XL Incorrect prediction. Actual species name is Corvus Albicollis. Section N.2.

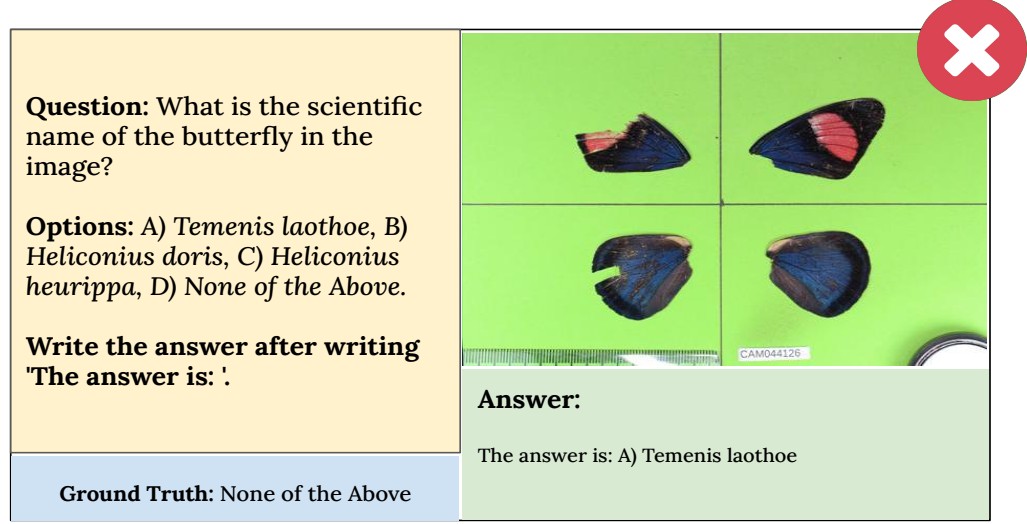

**Question:** What is the scientific name of the butterfly in the image?

**Options:** A) *Temenis laothoe*, B) *Heliconius doris*, C) *Heliconius heurippa*, D) *None of the Above.*

**Write the answer after writing 'The answer is: '.**

**Answer:**

The answer is: A) Temenis laothoe

**Ground Truth:** None of the Above

Figure 36: GPT-4o Incorrect prediction. Actual species name is *Batesia Hypochlora*. Section N.2.

**Question:** What is the scientific name of the butterfly in the image?

**Options:** A) *Heliconius erato*, B) *Heliconius melpomene*, C) *Heliconius eleuchia*, D) *None of the Above*.

**Write the answer after writing 'The answer is: '.**

**Ground Truth:** None of the Above

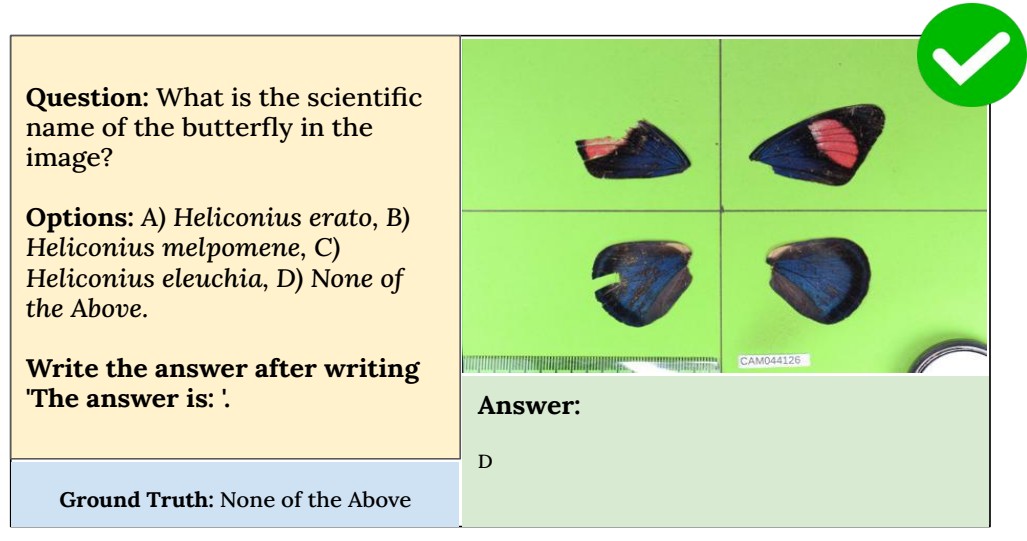

**Answer:**

D

Figure 37: Blip-Flan-XL Correct prediction. Actual species name is *Batesia Hypochlora*. Section N.2.

