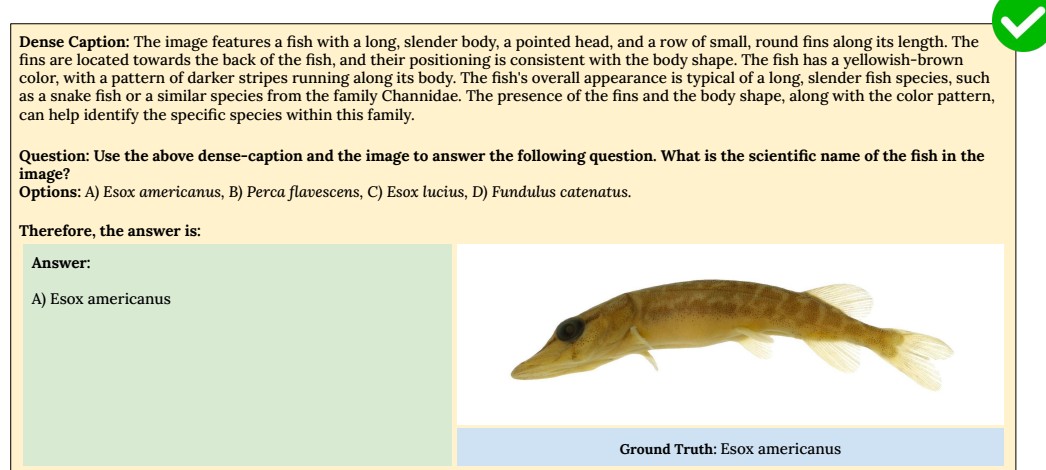

**Dense Caption:** The image features a fish with a long, slender body, a pointed head, and a row of small, round fins along its length. The fins are located towards the back of the fish, and their positioning is consistent with the body shape. The fish has a yellowish-brown color, with a pattern of darker stripes running along its body. The fish's overall appearance is typical of a long, slender fish species, such as a snake fish or a similar species from the family Channidae. The presence of the fins and the body shape, along with the color pattern, can help identify the specific species within this family.

**Question:** Use the above dense-caption and the image to answer the following question. What is the scientific name of the fish in the image?
**Options:** A) *Esox americanus*, B) *Perca flavescens*, C) *Esox lucius*, D) *Fundulus catenatus*.

**Therefore, the answer is:**

**Answer:**

A) Esox americanus

**Ground Truth:** Esox americanus

Figure 21: Dense Captions in Prompts. LLaVa-7B Correct prediction. Section M.3.

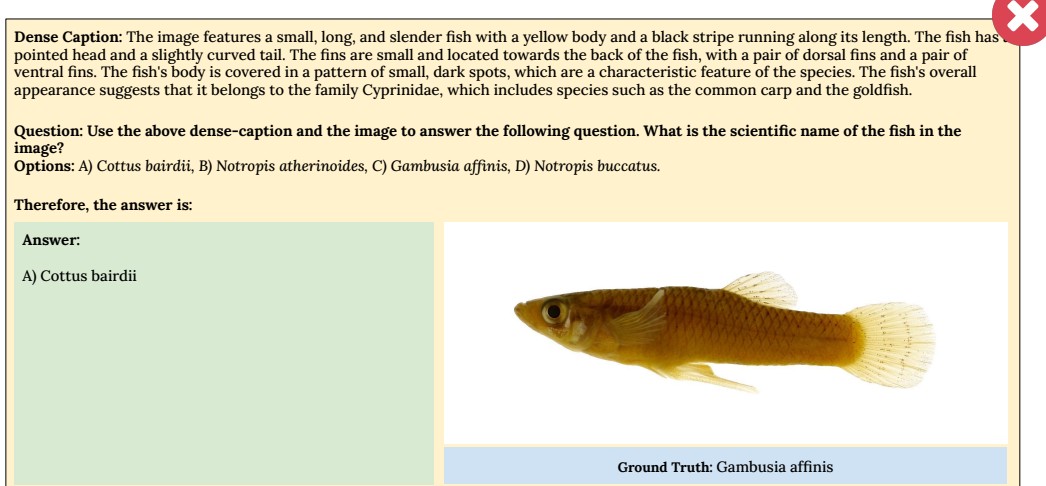

**Dense Caption:** The image features a small, long, and slender fish with a yellow body and a black stripe running along its length. The fish has a pointed head and a slightly curved tail. The fins are small and located towards the back of the fish, with a pair of dorsal fins and a pair of ventral fins. The fish's body is covered in a pattern of small, dark spots, which are a characteristic feature of the species. The fish's overall appearance suggests that it belongs to the family Cyprinidae, which includes species such as the common carp and the goldfish.

**Question:** Use the above dense-caption and the image to answer the following question. What is the scientific name of the fish in the image?
**Options:** A) *Cottus bairdii*, B) *Notropis atherinoides*, C) *Gambusia affinis*, D) *Notropis buccatus*.

**Therefore, the answer is:**

**Answer:**

A) Cottus bairdii

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

**Dense Caption:** The image features a fish with a long, slender body, a pointed head, and a row of small, round fins along its length. The fins are located towards the back of the fish, and their positioning is consistent with the body shape. The fish has a yellowish-brown color, with a pattern of darker stripes running along its body. The fish's overall appearance is typical of a long, slender fish species, such as a snake fish or a similar species from the family Channidae. The presence of the fins and the body shape, along with the color pattern, can help identify the specific species within this family.

**Question:** Use the above dense-caption and the image to answer the following question. What is the scientific name of the fish in the image?
**Options:** A) *Esox americanus*, B) *Perca flavescens*, C) *Esox lucius*, D) *Fundulus catenatus*.

**Therefore, the answer is:**

**Answer:**

A) Esox americanus

**Ground Truth:** Esox americanus

Figure 49: Dense Captions in Prompts. LLaVa-7B Correct prediction. Section M.3.

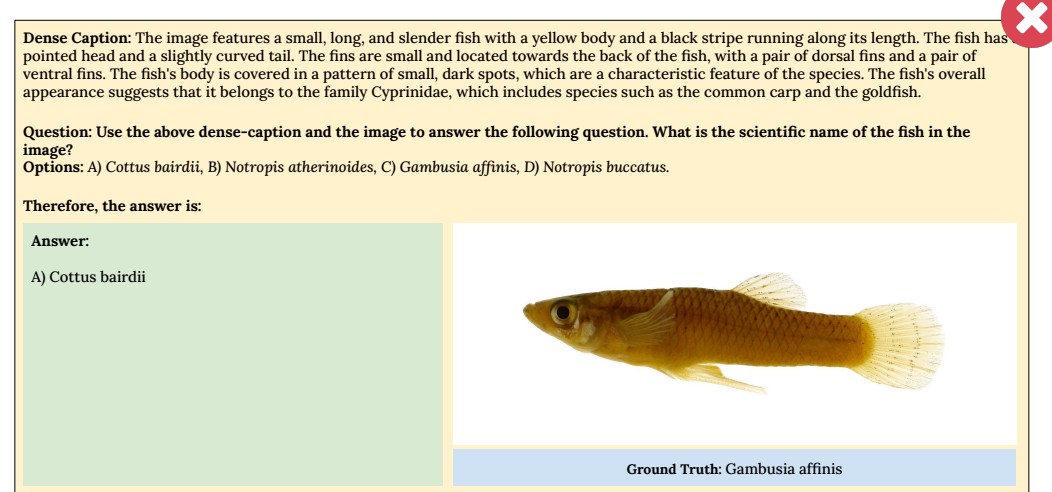

**Dense Caption:** The image features a small, long, and slender fish with a yellow body and a black stripe running along its length. The fish has pointed head and a slightly curved tail. The fins are small and located towards the back of the fish, with a pair of dorsal fins and a pair of ventral fins. The fish's body is covered in a pattern of small, dark spots, which are a characteristic feature of the species. The fish's overall appearance suggests that it belongs to the family Cyprinidae, which includes species such as the common carp and the goldfish.

**Question:** Use the above dense-caption and the image to answer the following question. What is the scientific name of the fish in the image?
**Options:** A) *Cottus bairdii*, B) *Notropis atherinoides*, C) *Gambusia affinis*, D) *Notropis buccatus*.

**Therefore, the answer is:**

**Answer:**

A) Cottus bairdii