# OpenReview forum: "VLM4Bio: A Benchmark Dataset to Evaluate Pretrained Vision-Language Models for Trait Discovery from Biological Images"
_NeurIPS.cc/2024/Datasets_and_Benchmarks_Track — NeurIPS 2024 Track Datasets and Benchmarks Poster_

### Official Review · Reviewer_8hpk · 2024-07-19
**Biodiversity dataset with extensive evaluation**

**Rating:** 6
**Confidence:** 3
**Correctness:** Image preprocessing steps, annotation…

**Review:**

The work presents a dataset\benchmark for organismal biology domain, which is quite uncommon. Authors put effort into annotations and VQA ground-truth. At the moment, the dataset has its limitations, i.e. covers only a few organisms, but hopefully that dataset would be expanded in the future, at the time this benchmark would have limited impact to the community. There are also some questions on dataset sampling that author could improve or clarify. The paper is clearly written, though lacks some details on how the source datasets were sampled.

**Strengths:**

**Relatively uncommon data domain and first benchmark on animal species**: it is good to see the datasets of different domains coming and benchmarked.
**Evaluation of the results**: authors evaluated many VLMs with their dataset and provided results for the baselines and additional validation:  * false confidence* and *none of the above*.
I hope the dataset would be expanded in future to cover more species and eventually make more impact for studying biodiversity.

**Additional Feedback:**

NA

**Clarity:**

The paper is clearly written, though lacks some details on how the source datasets were sampled.

**Documentation:**

There is a link to GitHub in abstract, the page is well-documented. Dataset is hosted on Hugging Face.

**Ethics:**

I see no ethical concerns.

**Limitations:**

Authors mention a limitation on number of organisms represented in the dataset.
Still, the current version of the dataset is good enough for the tasks it was designed.

**Opportunities For Improvement:**

- Lines 124-125: Fish-10k dataset was randomly sampled. I would expect more thorough selection, maybe random sampling after clustering of features extracted with a pretrained model (say, DINOv2 or even ImageNet pretrained ResNet) to ensure diversity and balance of this part of the dataset.
- Line 129: “carefully sampled 10K images”  - could you specify how exactly? I would suggest doing a similar procedure as for the Fish subset.
- Classes in Fish\Butterfly subsets could be more balanced.
- As an additional limitation I see that the bird subset is sampled only from one source.

**Relation To Prior Work:**

Related works section misses the discussion and references to at least image datasets and existing benchmarks of organismal biology. For instance. iNaturalist dataset or Species196.

**Summary And Contributions:**

The paper introduces a dataset from a relatively uncommon domain of organismal biology that includes species of fishes, birds and butterflies, covering five biologically relevant tasks. Authors evaluate zero-shot effectiveness of pretrained VLMs, study of prompting and VLM hallucinations. Authors employed GPT-4V, GPT-4o and Gemini and other (in total 12) VLMs for benchmarking.

---

> ### Author Rebuttal · Authors · 2024-08-17
>
> Thank you very much for your thorough review and appreciation of our work. Below, we address your concerns point by point:
>
> **Concern 1: Lines 124-125: Fish-10k dataset was randomly sampled. I expect a more thorough selection, maybe random sampling after clustering of features extracted with a pretrained model (say, DINOv2 or even ImageNet pretrained ResNet) to ensure diversity and balance of this part of the dataset.**
> >**Response:** Thank you for the insightful comment. To ensure diversity within the Fish-10K dataset, we applied a targeted sampling strategy in the source collection, FishAIR [1]. Specifically, we retained all images of species with fewer than 200 images, considering these as minority or rare classes. Random sampling was applied only to the majority species—those with more than 200 images per class. To assess the potential sampling bias among the majority species, we generated feature vectors for each image in Fish-10K using a pretrained VGG-19 model. In Figure 3 of the attached rebuttal document (pdf), we present species-wise t-SNE plots of these feature vectors for several majority species. Our analysis shows that the distribution of sampled images closely mirrors the distribution of images that were not included in the dataset (denoted as "others" in the plot). This suggests that our random sampling approach provides a sufficiently accurate representation of the original distribution for the majority species.
> >
> >*[1] ​​Multimedia of fish specimen and associated metadata. fish-air. Biology-guided Neural Network. Tulane University Biodiversity Research Institute, fishair.org.*
>
> **Concern 2: Line 129: “carefully sampled 10K images” - could you specify how exactly? I would suggest using a procedure similar to that for the Fish subset.**
> >**Response:** We apologize for the lack of detail in our original statement. Here are some additional details to clarify our sampling procedure. We sourced the images for the Butterfly-10K dataset from the Jiggins Heliconius collection [2]. These images show butterflies from various angles, including dorsal and ventral views, forewing dorsal and ventral views, and hindwing dorsal and ventral views. To ensure consistency, we only selected images with dorsal and removed all images of hybrid species. For species with more than 2000 images, we performed random sampling (no sampling was performed for species with sizes less than 2000). The Butterfly-10K dataset contains a significant number of images of 'Heliconius melpomene' and 'Heliconius erato' species. We utilized the subspecies information of these two species to create a hard dataset for analyzing the impact of answer choices on VLM performance, as described in Section 5.1. We will include these details in the final version of the Supplementary Material.
> >
> >*[2] Christopher Lawrence and Elizabeth G. Campolongo. Heliconius collection (Cambridge butterfly), 2024.*
>
> **Concern 3: Classes in Fish\Butterfly subsets could be more balanced.**
> >**Response**: The imbalance in Fish-10K and Butterfly-10K reflects the natural imbalance in the occurrence and observation of species in museum collections. Due to the scarcity of images for the rare species, it is difficult to increase their representation to avoid imbalance. As a result, we have included many under-represented species in the Fish and Butterfly datasets to report performance on the rare classes. For our experiments in sections 5.1, 5.3, and 5.4, we use balanced subsets of the VLM4Bio - easy, medium, hard, and prompting datasets. Additionally, the evaluation of VLM performance on a balanced dataset is demonstrated on the Bird-10K dataset.
>
>
> **Concern 4: The bird subset is sampled only from one source.**
> >**Response:** Thank you for this suggestion. Since the focus of our work is on understanding biological traits, we selected the CUB Bird-10K dataset, which contains classification labels as well as trait identification labels. Note that other available bird datasets like Bird-525 [3] are only focused on classification task. We will work on adding bird images from other sources in future versions of our dataset.
> >
> >[3] Gerald Piosenka. Birds 525 species - image classification. 05 2023.
>
> **Concern 5: The related works section misses the discussion and references to at least image datasets and existing benchmarks of organismal biology.**
> >**Response:** Thank you for this observation. In our related work, we mainly focus on existing benchmark datasets for evaluating VLMs on scientific tasks. Several benchmark datasets exist for organismal biology, such as iNaturalist, Species196, and BIOSCAN Insect 1M dataset, which primarily focus on the classification task. On the other hand, one of our primary goals is to evaluate VLMs on tasks involving biological traits going beyond classification, such as trait detection, identification, and segmentation, which are unavailable in these existing datasets. We will also include references to iNaturalist, Species196, and BIOSCAN Insect 1M in the revised draft and plan to compare VLMs on these datasets in future versions of our work.
>
> If you still have any concerns or aspects you would like to discuss further, please do not hesitate to contact us at any time.

---

> > ### Comment · Reviewer_8hpk · 2024-08-28
> >
> > Thank you for the answers, I don't have any additional questions and my review remains positive.

---

### Official Review · Reviewer_W944 · 2024-07-25
**Interesting benchmark**

**Rating:** 7
**Confidence:** 3
**Correctness:** Looks correct overall.
**Clarity:** The manuscript is clear.

**Review:**

Overall the paper looks good. The evaluation is extensive and considers various models and prompting strategies. In addition, the paper presents datasets that with various types of questions, images pre-processed for this task and region annotations.

As a downside, the images come from publicly available datasets that were curated before, most of the annotations are automatically generated, and some of the questions (e.g. is the eye visible) are trivial.

**Strengths:**

The experimental evaluation is extensive.

**Additional Feedback:**

None.

**Documentation:**

Mostly complete and well structured.

**Ethics:**

No concerns.

**Limitations:**

See above.

**Opportunities For Improvement:**

The paper is good enough and informative.

**Relation To Prior Work:**

Prior work is presented and discussed well.

**Summary And Contributions:**

The paper presents a dataset with biological images and visual question-answer pairs for evaluation of vision-language models. The manuscript also presents an extensive evaluation of existing models and reports performance.

---

> ### Author Rebuttal · Authors · 2024-08-17
>
> We are encouraged to see that you found our work informative, containing extensive experimental evaluation, and presented and discussed well. We have endeavored to address your concerns as follows.
>
> **Concern 1:  The images come from publicly available datasets; most of the annotations are automatically generated.**
> >**Response:** We agree with the reviewer that the images in the VLM4Bio are curated from previously available datasets, and most of the annotations were automatically generated. However, the motivation for this was to develop an organismal biology dataset in a VQA format that is readily usable for evaluating VLMs. Further, the automatic generation of questions allows us to curate a large dataset with 469K questions.
>
> **Concern 2:  Some of the questions (e.g., is the eye visible) are trivial.**
> >**Response:** The rationale behind including trait identification questions such as "Is the eye visible in the image?" is to enable a deeper comparison with trait grounding and referring tasks. While these questions may seem trivial, they are intentionally designed to understand the inner workings of VLMs. Specifically, we aim to determine whether a VLM answers these simple questions based on general knowledge that traits like eyes are typically present in organisms (e.g., fishes) or if it is capable of accurately localizing the eye within the image before determining its presence or absence. By comparing the results of these trait identification tasks with trait grounding/referring tasks, we aim to discover whether VLMs rely on visual cues to answer presence/absence questions. Also, VLM4Bio involves questions with varying levels of difficulty. While there are some easier questions questioning the presence of an eye or head in fishes, there are also harder questions regarding bill color, wing pattern, and tail shape, requiring a thorough understanding of the visual cues present in the image. We have shown the different traits used in the identification tasks in Figure 6 in the Supplementary for reference.
>
> If you still have any concerns or aspects you would like to discuss further, please do not hesitate to contact us at any time.

---

> > ### Comment · Reviewer_W944 · 2024-08-27
> >
> > Thank you for the comments. I don't have additional questions at this time.

---

### Official Review · Reviewer_eVWa · 2024-08-07

**Rating:** 7
**Confidence:** 4

**Review:**

- quality: good. As a benchmark work, this paper provide a reasonable data collection process, five useful tasks to test the biologically related capabilities, and comprehensive evaluations.
  - clarity: This paper was clearly written and easy to follow.
  - originality: novel.
  - significance: The proposed dataset is based on existing data corpus that have been used to evaluate the traditional biological related models, and this paper convert and annotate these data source into the VQA format to assess modern VLMs. The significance is relatively trivial.

**Strengths:**

- The newly proposed dataset that contains 469K QA pairs and 30K images in the organismal biological domain is useful and promising.
  - The evaluation for current up-to-date VLMs is insightful, that can validate the VLMs' capabilities on the specific biological domain.

**Additional Feedback:**

Will the backgrounds of the images that you have removed based on SAM affect the performance of VLMs?

**Clarity:**

This paper is generally well written. There are several typos:

  - At line 29, LLaMA is not a vision-language model.

**Correctness:**

The claims made in this paper are correct. The constructed dataset is in a sound way. The evaluation process is performed correctly.

**Documentation:**

This paper lacks a description of the license.

**Ethics:**

No ethical concerns.

**Limitations:**

- It is well known that VLMs can achieve commended performance on those images that were seen during training, and poor performance on identifying OOD images. Therefore the zero-shot evaluation is not very insightful as most of VLMs are trained on datasets centered around human activities. It would be more convincing to fine-tune VLMs on related biological datasets and then validate them on the five tasks.
  - I think the resolution of the input image would be an important factor to affect the VLMs performance on recognizing the biological details. However, there are not experiments demonstrating this impacts.

**Opportunities For Improvement:**

- For the Trait Grounding case in Figure 3, there should be the corresponding annotations on the images for the bounding boxes in the Ground Truth for a smooth understanding.

**Relation To Prior Work:**

Yes.

**Summary And Contributions:**

This paper aims at assessing the zero-shot capabilities of existing VLMs on answering a range of organismal biological questions, including the tasks of species classification, trait identification, trait grounding, trait referring, and trait counting, based on a newly proposed dataset called VLM4Bio. Their main contribution are as follows:
  - A new dataset that contains 469K QA pairs and 30K images is proposed to evaluate the performance of VLMs.
  - A comprehensive evaluation for 12 SOTA VLMs.

---

> ### Author Rebuttal · Authors · 2024-08-17
>
> We thank you for the positive comments on the novelty and meaningful impact of our findings and proposed benchmark. We detail your concerns and our corresponding responses below:
>
> **Concern 1: Trait annotation on the images in Figure 3 for smooth understanding.**
> >**Response:** Thank you for this suggestion. Please refer to Figure 2 of the attached rebuttal document (pdf), where we have modified the figure accordingly. We will update Figure 3 in the final version of the main paper.
>
> **Concern 2: The zero-shot evaluation is not very insightful, as most VLMs are trained on datasets centered around human activities. It would be more convincing to fine-tune VLMs on related biological datasets and then validate them on the five tasks.**
> >**Response:** We agree with the reviewer that fine-tuning VLMs on biological datasets can improve their performance on the five tasks, which indeed require investigation and validation and is currently outside the scope of this work. Zero-shot evaluation benchmarks are crucial for continually assessing the capabilities of Vision-and-Language Models (VLMs). For example, the MMMU benchmark, introduced in November 2023, is a valuable tool for evaluating the zero-shot capabilities of newly developed VLMs across diverse fields such as Art, Design, Business, Science (including biology), Health, Medicine, and Engineering. However, the MMMU benchmark does not cover organismal biology, which is the primary focus of our work. Currently, there is no benchmark dataset specifically designed to assess visual trait understanding in organismal biology. To fill this gap, we are introducing the VLM4Bio benchmark, which will allow for the ongoing evaluation of future VLMs in the context of organismal biology. The VLM4Bio benchmark will enable biologists to assess the effectiveness of advanced VLMs in addressing scientific questions across various biologically relevant tasks within the field of organismal biology.
>
>
> **Concern 3: Resolution of the input image would be an important factor to affect the VLMs performance on recognizing the biological details.**
> >**Response:** Thank you for this comment. To investigate the effect of image resolution on VLM performance, we performed additional experiments summarized in Figure 1 of the attached rebuttal document. In this Figure, we show distribution plots for the Fish-10K and Bird-10K datasets with variations in the image resolutions and their impact on the species classification performance (MC question format) for GPT-4V, LLaVA-1.5-7B, and LLaVA-1.5-13B. All the images of the Butterfly-10K have the exact resolution (500x333); hence, they were not included in the experiment.
> >From Figure 1(c), it is clear that image resolution is influential on the VLM performance for the Fish-10K dataset since higher resolution helps in recognizing the details of the biological traits and correct species. However, for Figure 1(d), the VLM performances do not vary significantly with the image resolution for the Bird-10K dataset. A potential reason is that the bird dataset is a subset of the CUB dataset, and we hypothesize that the pre-trained VLMs may have seen images with resolutions similar to those in the Bird-10K dataset during training, leading to this behavior.
>
> **Concern 4: Will the backgrounds of the images that you have removed based on SAM affect the performance of VLMs?**
> >**Response:** Yes. The Fish-10K dataset contains images of specimens preserved in museum collections with artificial backgrounds with imaging artifacts that are not typical for large-scale computer vision datasets. Moreover, these backgrounds can introduce unexpected bias. Hence, we removed the backgrounds using SAM to create a controlled environment for our experiments.
>
> **Concern 5: Description of the license.**
> >**Response:** We have provided details of the dataset Licenses in Supplementary Section E. The image-specific licenses have been provided in the HuggingFace repository. For the ease of the readers, we have modified the README file to highlight the license section.
>
> Your constructive comments will greatly assist us in improving this work. Please do not hesitate to contact us if you have any further questions.

---

> > ### Comment · Reviewer_eVWa · 2024-08-29
> >
> > Thank you for your response and additional experiments. These improvements are meaningful and should be included in the final version of the paper. I have raised my rating accordingly.

---

### Author Rebuttal · Authors · 2024-08-17

We sincerely appreciate all reviewers for their time and efforts in the review. Their insightful comments and constructive feedback have helped us refine the quality of our work. All detailed questions of each reviewer are answered accordingly in each column below. We have attached a pdf to include the associated figures and tables. We hope these responses address the reviewers' concerns adequately.

We are encouraged that the reviewers found our work:
- Well-written and easy to follow. (Reviewers *eVWa, W944, 8hpk*)
- Useful and promising benchmark for studying biodiversity. (Reviewers *eVWa, 8hpk*)
- The experimental evaluation is extensive. (Reviewers *eVWa, W944, 8hpk*)
- Insightful evaluation for up-to-date VLMs validating their capabilities in the biological domain. (Reviewers *W944,8hpk*)

We take the opportunity to re-emphasize the **main contribution** of this paper and its impact on the scientific community:
1. We present a novel dataset of scientific question-answer pairs to evaluate the effectiveness of VLMs in answering scientific questions across a range of biologically relevant tasks in the field of organismal biology.
2. We present novel benchmarking analyses of the zero-shot effectiveness of pre-trained SOTA VLMs on our dataset, exposing their gaps in advancing scientific knowledge of organismal biology.
3. We present novel comparisons studying the effects of prompting and tests for reasoning hallucination on VLM performance, shedding new light on the reasoning capabilities of SOTA VLMs in organismal biology.

Even though we focus on three organisms currently, we are able to provide new insights into zero-shot performance and the limitations of the VLMs. Adding more organism datasets could enhance the insights that we plan to explore in the future.

We have provided our response to each reviewer's comments separately below. Here is a summary of the main comments raised by the reviewers and the revisions we have made to address them:

1. **Comment:** More details of data sampling (Reviewers *8hpk*) **Response:** We have added more details about the sampling process and provided additional visualizations in Figure 3 of the rebuttal document (pdf) to demonstrate the absence of sampling bias.
2. **Comment:** Impact on image size (Reviewers *eVWa*): **Response:** We have provided experimental results in Figure 1 of the rebuttal document to demonstrate that resolution helps in the recognition of biological traits.

---

> ### Author Response · Authors · 2024-08-27
> **Closing Remarks and Request for Final Reviewer Comments and Score Updates**
>
> We sincerely appreciate the insightful comments and valuable feedback provided by our reviewers. As we approach the conclusion of the discussion period, we would like to extend an invitation for any further questions or clarifications. If our responses have adequately addressed your concerns, we kindly request that you consider revising your scores accordingly. Thank you for your time and dedication to this review process.

---

### Decision · Program_Chairs · 2024-09-26

**Decision:**

Accept (Poster)

**Comment:**

The paper introduces biology dataset consisting of Q&A pairs for fishes, birds and butterflies, coined as VLM4Bio, and evaluates SOTA VLMs on it.

Strengths identified by reviewers:
* The paper is well-written and easy to follow.
* The proposed VLM4Bio dataset is useful and promising for studying biodiversity.
* The paper provides offers an extensive evaluation and insightful validation of the capabilities of up-to-date VLMs in the biological domain.

The reviewers also pointed out some weaknesses:

* The zero-shot evaluation might not be very insightful as most VLMs are trained on datasets centered around human activities (Reviewer eVWa).
* The impact of image resolution on VLM performance was not initially explored (Reviewer eVWa) To address this, the authors conducted additional experiments to demonstrate the impact of image resolution on VLM performance.
* Some of the questions in the dataset might be considered trivial (Reviewer W944). Here the authors note that the goal is to see if the VLM can localize the answer in the image.
* The dataset currently covers only a few organisms (Reviewer 8hpk).


Overall I think the paper meets the bar of publication and recommend acceptance.